# Diffusion-DICE: In-Sample Diffusion Guidance for Offline Reinforcement Learning

**Liyuan Mao**[*]
Shanghai Jiao Tong University

**Haoran Xu**[*]
UT Austin

**Xianyuan Zhan**
Tsinghua University

**Weinan Zhang**[†]
Shanghai Jiao Tong University

**Amy Zhang**[†]
UT Austin

## Abstract

One important property of DIstribution Correction Estimation (DICE) methods is that the solution is the optimal stationary distribution ratio between the optimized and data collection policy. In this work, we show that DICE-based methods can be viewed as a transformation from the behavior distribution to the optimal policy distribution. Based on this, we propose a novel approach, Diffusion-DICE, that directly performs this transformation using diffusion models. We find that the optimal policy's score function can be decomposed into two terms: the behavior policy's score function and the gradient of a guidance term which depends on the optimal distribution ratio. The first term can be obtained from a diffusion model trained on the dataset and we propose an in-sample learning objective to learn the second term. Due to the multi-modality contained in the optimal policy distribution, the transformation in Diffusion-DICE may guide towards those local-optimal modes. We thus generate a few candidate actions and carefully select from them to approach global-optimum. Different from all other diffusion-based offline RL methods, the *guide-then-select* paradigm in Diffusion-DICE only uses in-sample actions for training and brings minimal error exploitation in the value function. We use a didatic toycase example to show how previous diffusion-based methods fail to generate optimal actions due to leveraging these errors and how Diffusion-DICE successfully avoids that. We then conduct extensive experiments on benchmark datasets to show the strong performance of Diffusion-DICE.

## 1 Introduction

We study the problem of offline reinforcement learning (RL), where the goal is to learn effective policies solely from offline data, without any additional online interactions. Offline RL is quite useful for scenarios where arbitrary exploration with untrained policies is costly or dangerous, but sufficient prior data is available, such as robotics [18] or industrial control [56]. Most previous model-free offline RL methods add a pessimism term to off-policy RL algorithms [50, 9, 2], this pessimism term acts as behavior regularization to avoid extrapolation errors caused by querying the $Q$-function about values of potential out-of-distribution (OOD) actions produced by the policy [25]. However, explicitly adding the regularization requires careful tuning of the regularization weight because otherwise, the policy will still output actions that are not seen in the dataset. DIstribution Correction Estimation (DICE) methods [35, 41, 34] provide an implicit way of doing so. By applying convex duality, DICE-based methods solve for the optimal stationary distribution ratio between the optimized and data collection policy in an in-sample manner without querying unseen actions [55].

---

[*]Equal contribution; more junior authors listed earlier.
[†]Corresponding authors. Project page at `https://ryanxhr.github.io/Diffusion-DICE/`.

38th Conference on Neural Information Processing Systems (NeurIPS 2024).

In this paper, we extend the analysis of DICE-based methods: we show that the optimal distribution ratio in DICE-based methods can be extended to a transformation from the behavior distribution to the optimal policy distribution. This different view motivates the use of deep generative models, e.g. diffusion models [42, 15, 43], to first fit the behavior distribution using their strong expressivity of fitting multi-modal distributions and then directly perform this transformation during sampling. We find that the optimal policy's score function can be decomposed into two terms: the behavior policy's score function and the gradient of a guidance term which depends on the optimal distribution ratio learned by DICE. The first term can be easily obtained from the diffusion model trained on the offline dataset. While it is usually intractable to find a closed-form solution of the second term, we make a subtle mathematical transformation and show its equivalence to solving a convex optimization problem. In this manner, both of these terms can be learned by only dataset samples, providing accurate guidance towards in-distribution while high-value data points. Due to the multi-modality contained in the optimal policy distribution, the transformation may guide towards those local-optimal modes due to stochasticity. We thus generate a few candidate actions and use the value function to select the max from them to go towards the global optimum.

We term our method Diffusion-DICE, the *guide-then-select* procedure in Diffusion-DICE differs from all previous diffusion-based offline RL methods [49, 4, 14, 32], which are either only guide-based or only select-based. Guide-based methods [32] use predicted values of actions generated by the diffusion behavior policy to guide toward high-return actions. Select-based methods [4, 14] bypass the need for guidance but require sampling a large number of actions from the diffusion behavior policy and using the value function to select the optimal one. All these methods need to query the value function

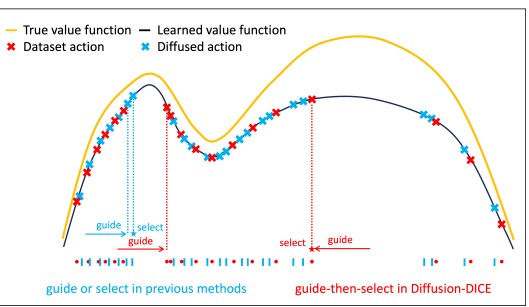

Figure 1: Illustration of the *guide-then-select* paradigm

of actions sampled from the diffusion behavior policy, which may produce potential OOD actions and bring overestimation errors in the guiding or selecting process. Our method, however, brings minimal error in the guide-step by using accurate in-sample guidance to generate in-distribution actions with high values, and by doing so, in the select step only a few candidate actions are needed to find the optimal one, which further reduces error exploitation. We use an illustrative toy case to demonstrate the error exploitation in previous methods and how Diffusion-DICE successfully alleviates that. We also verify the effectiveness of Diffusion-DICE in benchmark D4RL offline datasets [8]. Note that Diffusion-DICE also provides a replacement for the Gaussian policy extraction part used in current DICE methods, successfully unleashing the power of DICE-based methods. Diffusion-DICE surpasses both diffusion-based and DICE-based strong baselines, reaching SOTA performance in D4RL benchmarks [8]. We also conduct ablation experiments and validate the superiority of the guide-then-select learning procedure.

## 2 Preliminaries

We consider the RL problem presented as a Markov decision process [45], which is specified by a tuple $\mathcal{M} = \langle \mathcal{S}, \mathcal{A}, \mathcal{P}, d_0, r, \gamma \rangle$. Here $\mathcal{S}$ and $\mathcal{A}$ are state and action space, $\mathcal{P}(s'|s,a)$ and $d_0$ denote transition dynamics and initial state distribution, $r(s,a)$ and $\gamma$ represent reward function and discount factor, respectively. The goal of RL is to find a policy $\pi(a|s)$ which maximizes expected return $\mathbb{E}[\sum_{t=0}^{\infty} \gamma^t \cdot r(s_t, a_t)]$. Another equivalent LP form of expected return is $\mathbb{E}_{(s,a) \sim d^\pi}[r(s,a)]$, where $d^\pi(s,a) := (1-\gamma) \sum_{t=0}^{\infty} \gamma^t Pr(s_t = s, a_t = a)$ is the *normalized discounted stationary distribution* of $\pi$ [36]. For simplicity, we refer to $d^\pi(s,a)$ as the stationary distribution of $\pi$. Offline RL considers the setting where interaction with the environment is prohibited, and one need to learn the optimal $\pi$ from a static dataset $\mathcal{D} = \{s_i, a_i, r_i, s_i'\}_{i=1}^N$. We denote the empirical behavior policy of $\mathcal{D}$ as $\pi^{\mathcal{D}}$.

### 2.1 Distribution Correction Estimation

DICE methods [41] incorporate the LP form of expected return $\mathcal{J}(\pi) = \mathbb{E}_{(s,a) \sim d^\pi}[r(s,a)]$ with a regularizer $D_f(d^\pi || d^{\mathcal{D}}) = \mathbb{E}_{(s,a) \sim d^{\mathcal{D}}}[f(\frac{d^\pi(s,a)}{d^{\mathcal{D}}(s,a)})]$, where $D_f$ is the $f$-divergence induced by a

convex function $f$ [3]. More specifically, DICE methods try to find an optimal policy $\pi^*$ that satisfies:

$$\pi^* = \arg\max_{\pi} \mathbb{E}_{(s,a)\sim d^\pi}[r(s,a)] - \alpha D_f(d^\pi || d^{\mathcal{D}}). \tag{1}$$

This objective is generally intractable due to the dependency on $d^\pi(s,a)$, especially under the offline setting. However, by imposing the *Bellman-flow* constraint [33] $\sum_{a \in \mathcal{A}} d(s,a) = (1-\gamma)d_0(s) + \gamma \sum_{(s',a')} d(s',a')p(s|s',a')$ on states and applying Lagrangian duality and convex conjugate, its dual problem has the following tractable form:

$$\min_{V} (1-\gamma)\mathbb{E}_{s\sim d_0}[V(s)] + \alpha\mathbb{E}_{(s,a)\sim d^{\mathcal{D}}}[f^*([\mathcal{T}V(s,a) - V(s)]/\alpha)]. \tag{2}$$

Here $f^*$ is a variant of $f$'s convex conjugate and $\mathcal{T}V(s,a) = r(s,a) + \gamma\mathbb{E}_{s'\sim p(\cdot|s,a)}[V(s')]$ represents the Bellman operator on $V$. In practice, one often uses a prevalent semi-gradient technique in RL that estimates $\mathcal{T}V(s,a)$ with $Q(s,a)$ and replaces the initial state distribution $d_0$ with dataset distribution $d^{\mathcal{D}}$ to stabilize learning [41, 34]. In addition, because $\mathcal{D}$ usually cannot cover all possible $s'$ for a specific $(s,a)$, DICE methods only use a single sample of the next state $s'$. The update of $Q(s,a)$ and $V(s)$ in DICE methods are as follows and we refer to a detailed derivation in Appendix A:

$$\min_{V} \mathbb{E}_{(s,a)\sim d^{\mathcal{D}}}\left[(1-\gamma)V(s) + \alpha f^*\left([Q(s,a) - V(s)]/\alpha\right)\right]$$
$$\min_{Q} \mathbb{E}_{(s,a,s')\sim d^{\mathcal{D}}}\left[\left(r(s,a) + \gamma V(s') - Q(s,a)\right)^2\right]. \tag{3}$$

Note that learning objectives of DICE-methods can be calculated solely with a $(s,a,s')$ sample from $\mathcal{D}$, which is totally in-sample. One important property of DICE methods is that $Q^*$ and $V^*$ have a relationship with the optimal stationary distribution ratio $w^*(s,a)$ as

$$w^*(s,a) := \frac{d^*(s,a)}{d^{\mathcal{D}}(s,a)} = \max\left(0, (f')^{-1}\left(Q^*(s,a) - V^*(s)\right)\right), \tag{4}$$

where $d^*$ is the stationary distribution of $\pi^*$. To get $\pi^*$ from $w^*$, previous policy extraction methods in DICE methods include weighted behavior cloning [34], information projection [27] or policy gradient [37]. All these methods parametrize an unimodal Gaussian policy in order to compute $\log\pi(a|s)$ [13], which greatly limits its expressivity.

## 2.2 Diffusion Models in Offline Reinforcement Learning

Diffusion models [42, 15, 43] are generative models based on a Markovian noising and denoising process. Given a random variable $x_0$ and its corresponding probability distribution $q_0(x_0)$, the diffusion model defines a forward process that gradually adds Gaussian noise to the samples from $x_0$ to $x_T(T > 0)$. Kingma et al. [20] shows there exists a stochastic process that has the same transition distribution $q_{t0}(x_t|x_0)$ and Song et al. [43] shows that under some conditions, this process has an equivalent reverse process from $T$ to 0. The forward process and the equivalent reverse process can be characterized as follows, where $\bar{w}_t$ is a standard Wiener process in the reverse time.

$$q_{t0}(x_t|x_0) = \mathcal{N}(x_t|\alpha_t x_0, \sigma_t^2 \mathbf{I}) \quad dx_t = [f(t)x_t - g^2(t)\nabla_{x_t}\log q_t(x_t)]dt + g(t)d\bar{w}_t, \ x_T \sim q_T(x_T). \tag{5}$$

Here we slightly abuse the subscript $t$ to denote the diffusion timestep. $\alpha_t, \sigma_t$ are the noise schedule and $f(t), g(t)$ can be derived from $\alpha_t, \sigma_t$ [31]. For simplicity, we denote $q_{t0}(x_t|x_0)$ and $p_{0t}(x_0|x_t)$ as $q(x_t|x_0)$ and $p(x_0|x_t)$, respectively. To sample from $q_0(x_0)$ by following the reverse stochastic differential equation (SDE), the score function $\nabla_{x_t}\log q_t(x_t)$ is required. Typically, diffusion models use denoising score matching to train a neural network $\epsilon_\theta(x_t, t)$ that estimates the score function [46, 15, 43], by minimizing $\mathbb{E}_{t\sim\mathcal{U}(0,T), x_0\sim q_0(x_0), \epsilon\sim\mathcal{N}(\mathbf{0},\mathbf{I})}[\|\epsilon_\theta(x_t, t) - \epsilon\|^2]$, where $x_t = \alpha_t x_0 + \sigma_t\epsilon$. As we mainly focus on diffusion policy in RL, this objective is usually impractical because $q_0(x_0)$ is expected to be the optimal policy $\pi^*(a|s)$. A more detailed discussion is given in Appendix A.

To make diffusion models compatible with RL, there are generally two approaches: *guide-based* and *select-based*. Guide-based methods [32, 17] incorporate the behavior policy's score function with an additional guidance term. Specifically, they learn a time-dependent guidance term $\mathcal{J}_t$ and use it to drift the generated actions towards high-value regions. The learning objective of $\mathcal{J}_t$ can be generally formalized with $\mathcal{L}(\mathcal{J}_t(a_t), w(s, \{a_0^i\}_{i=1}^K))$, where $w(s, \{a_0^i\}_{i=1}^K)$ are critic-computed values on $K$ diffusion behavior samples. $\mathcal{L}$ can be a contrastive objective [32] or mean-square-error objective [17].

After training, the augmented score function $\nabla_{a_t} \log \pi_t(a_t|s) = \nabla_{a_t} \log \pi_t^{\mathcal{D}}(a_t|s) + \nabla_{a_t} \mathcal{J}_t(a_t)$ is used to characterize the learned policy.

Select-based methods [4, 14] utilize the observation that for some RL algorithms, the actor induced through critic learning manifests as a reweighted behavior policy. To sample from the optimized policy, these methods first sample multiple candidates $\{a_0^i\}_{i=1}^N$ from the diffusion behavior policy and then resample from them using critic-computed values $w(s, \{a_0^i\}_{i=1}^N)$. More precisely, the sampling procedure follows the categorical distribution $Pr[a = a_0^j|s] = \frac{w(s,a_0^j)}{\sum_{i=1}^N w(s,a_0^i)}$.

**Error exploitation**   As we can see, both guide-based and select-based methods need the information of $w(s, \{a_0^i\}_{i=1}^N)$ to get the optimal action. However, this term may bring two sources of errors. One is the diffusion model's approximation error in modeling complicated policy distribution, and the other is the critic's error in evaluating unseen actions. Although trained on offline data, the diffusion model may still generate OOD actions (especially with frequent sampling) and the learned critic can make erroneous predictions on these OOD actions, causing the evaluated value on these actions to be over-estimated due to the learning nature of value functions [11, 25]. As a result, the generation of high-quality actions in existing methods is greatly affected due to this error exploitation, which we will also show empirically in the next section.

## 3   Diffusion-DICE

In this section, we introduce our method, Diffusion-DICE. We start from an extension of DICE-based method that considers DICE as a transformation from the behavior distribution to the optimal policy distribution, which motivates us to use diffusion models to perform such transformation. We then propose a *guide-then-select* procedure to achieve the best action, i.e., we propose in-sample guidance learning for accurate policy transformation and use the critic to do optimal action selection to boost the performance. We also propose a piecewise $f$-divergence to stabilize the gradient during learning. We give an illustration of the guide-then-select paradigm and use a toycase to showcase the error exploitation in previous methods and how Diffusion-DICE successfully alleviates that.

### 3.1   An Optimal Policy Transformation View of DICE

As mentioned before, DIstribution Correction Estimation (DICE) methods, an important line of work in offline RL and IL provides us with an elegant way to estimate the optimal stationary distribution ratio between $d^*(s, a))$ and $d^{\mathcal{D}}(s, a)$ [28, 41, 34]. We show that this ratio also directly indicates a proportional relationship between the optimal in-support policy $\pi^*(a|s)$ and the behavior policy $\pi^{\mathcal{D}}(a|s)$. This proportional relationship enables us to transform $\pi^{\mathcal{D}}$ into $\pi^*$.

Our key observation is that the definition of $d^\pi(s, a)$ inherently reveals a bijection between $\pi(a|s)$ and $d^\pi(s, a)$. Given a relationship between $d^*(s, a)$ and $d^{\mathcal{D}}(s, a)$, we can use this bijection to derive a relationship between $\pi^*(a|s)$ and $\pi^{\mathcal{D}}(a|s)$. We formalize the bijection and derived relationship as:

$$\pi(a|s) = \frac{d^\pi(s, a)}{\int_{a \in \mathcal{A}} d^\pi(s, a)da}, \quad \pi^*(a|s) = \frac{\frac{d^*(s,a)}{d^{\mathcal{D}}(s,a)}}{\int_{a \in \mathcal{A}} \frac{d^*(s,a)}{d^{\mathcal{D}}(s,a)} \pi^{\mathcal{D}}(a|s)da} \pi^{\mathcal{D}}(a|s).$$

The proof is given in Appendix B. Since the denominator in the second equation involves an integral over $a \in \mathcal{A}$ and is unrelated to the chosen action, the relationship indicates $\pi^*(a|s) \propto \frac{d^*(s,a)}{d^{\mathcal{D}}(s,a)} \pi^{\mathcal{D}}(a|s)$. This means that the transformation between $d^{\mathcal{D}}(s, a)$ and $d^*(s, a)$ can be extended to the transformation between $\pi^{\mathcal{D}}(a|s)$ and $\pi^*(a|s)$. This transformation motivates the use of deep generative models, e.g. diffusion models, to first fit the behavior distribution using their strong expressiveness and then directly perform this transformation during sampling. More specifically, the score function of the optimal policy and the behavior policy satisfy the following relationship:

$$\nabla_{a_t} \log \pi_t^*(a_t|s) = \nabla_{a_t} \log \pi_t^{\mathcal{D}}(a_t|s) + \nabla_{a_t} \log \mathbb{E}_{a_0 \sim \pi^{\mathcal{D}}(a_0|a_t,s)}\left[\frac{d^*(s, a_0)}{d^{\mathcal{D}}(s, a_0)}\right]. \tag{6}$$

The proof is given in Appendix B. Intuitively, $\nabla_{a_t} \log \pi_t^{\mathcal{D}}(a_t|s)$ tells us how to generate actions from $\pi^{\mathcal{D}}$ and $\nabla_{a_t} \log \mathbb{E}_{\pi^{\mathcal{D}}(a_0|a_t,s)}[w^*(s, a_0)]$ tells us how to transform from $\pi^{\mathcal{D}}$ to in-support $\pi^*$ during the reverse diffusion process. As mentioned before, when representing $\pi^*$ with a diffusion model,

all we need is its score function $\nabla_{a_t} \log \pi_t^*(a_t|s)$. Equivalently, we focus on the right-hand side of Eq.(6). The first term is just the score function of $\pi^{\mathcal{D}}$, which is fairly easy to obtain from the offline dataset [14, 4]. To perform the transformation, we still require the second term. Fortunately, DICE allows us to acquire the inner ratio term in an in-sample manner directly from the offline dataset, and we will show in the next section how to exactly compute the whole second term using offline dataset.

## 3.2 In-sample Guidance Learning for Accurate Policy Transformation

Although DICE provides us with the optimal stationary distribution ratio, which is the cornerstone of the transformation, the second term on the right-hand side of Eq.(6) is still intractable due to the conditional expectation. Our key observation is that for arbitrary non-negative function $f(x)$, the optimizer of the following convex problem is unique and takes the desired form of log-expectation.

**Lemma 1.** *Given a random variable $X$ and its corresponding distribution $P(X)$, for any non-negative function $f(x)$, the following problem is convex and its optimizer is given by $y^* = \log \mathbb{E}_{x \sim P(X)}[f(x)]$,*

$$\min_y \mathbb{E}_{x \sim P(X)}[f(x) \cdot e^{-y} + y].$$

It is evident that the optimizer can be used to derive the second term in Eq.(6). In fact, we show the second term can be obtained directly from the offline dataset, if we optimize the following objective. Here $g_\theta$ is a guidance network parameterized by $\theta$ and we denote $\frac{d^*(s,a)}{d^{\mathcal{D}}(s,a)}$ as $w^*(s,a)$ for shorthand.

**Theorem 1.** $\nabla_{a_t} \log \mathbb{E}_{\pi^{\mathcal{D}}(a_0|a_t,s)}[w^*(s,a_0)]$ *can be obtained by solving the following optimization problem:*

$$\min_\theta \mathbb{E}_{t \sim \mathcal{U}(0,T)} \mathbb{E}_{a \sim \pi^{\mathcal{D}}(a|s)} \mathbb{E}_{a_t \sim p(a_t|a_0)} \left[ w^*(s,a) e^{-g_\theta(s,a_t,t)} + g_\theta(s,a_t,t) \right], \tag{7}$$

*as the optimal solution $\theta^*$ satisfies $\nabla_{a_t} g_{\theta^*}(s,a_t,t) = \nabla_{a_t} \log \mathbb{E}_{\pi^{\mathcal{D}}(a_0|a_t,s)}[w(s,a_0)]$.*

This objective has two appealing properties. Firstly, compared to guide-based and select-based methods, we don't need to use multiple actions generated by the behavior diffusion model, we only need one sample from $\pi^{\mathcal{D}}(a|s)$ to estimate the second expectation, enabling an unbiased estimator of this objective using only offline dataset possible. Secondly, because $w(s,a)$ must be non-negative to ensure a valid transformation, this objective is convex with respect to $g_\theta(s,a_t,t)$. This indicates a guarantee of convergence to $g_{\theta^*}(s,a_t,t)$ under mild assumptions (see proof in Appendix B).

Directly inducing $\nabla_{a_t} \log \mathbb{E}_{\pi^{\mathcal{D}}(a_0|a_t,s)}[w^*(s,a_0)]$ from the dataset is crucial in the offline setting. Firstly, it avoids the evaluation of OOD actions and solely depends on reliable value predictions of in-sample actions. Consequently, the gradient exploits minimal error in the critic, enabling it to more accurately transform $\pi^{\mathcal{D}}$ into in-support $\pi^*$. Moreover, because all values of $w^*(s,a)$ used in our method are based on in-sample actions, applying this gradient in the reverse process will guide the samples towards in-sample actions with high $w^*(s,a)$ value.

We refer to this method as In-sample Guidance Learning (IGL). Note that previous methods either have a biased estimate of the gradient [17, 6], or have to rely on value predictions for diffusion generated actions, which could be OOD [32]. We refer to Appendix C for an in-depth discussion of IGL against other guidance methods.

**Stablizing gradient using piecewise $f$-divergence.** In practice, there exists one issue when computing the guidance term. Note that in Eq.(4), $w^*(s,a)$ could become zero, depends on the choice of $f$. Given a specific $a_t$, if $w^*(s,a_0) = 0$ for actions under $\pi^{\mathcal{D}}(a_0|a_t,s)$, the second gradient term will not be well-defined due to taking the logarithm of 0. In practice, this can result in an unstable gradient. To avoid such issue, $(f')^{-1}(x)$ should be positive for any given $x$. Also, to facilitate the optimization process, $f^*(x)$ should possess a closed-form solution and good numerical properties. Unfortunately, none of the commonly used $f$-divergence can accommodate both. However, considering the following two $f$-divergences:

| Divergence | $f(x)$ | $f^*(x)$ | $(f')^{-1}(x)$ |
|---|---|---|---|
| Reverse KL | $x \log x$ | $e^{x-1}$ | $e^{x-1}$ |
| Pearson $\chi^2$ | $(x-1)^2$ | $\max(-1, \frac{x^2}{4} + x)$ | $\max(0, \frac{x}{2} + 1)$ |

It is obvious that Reverse KL divergence possesses the positive property but exhibits numerical instability given a large $x$ due to the exponential function in $f^*(x)$, while Pearson $\chi^2$ divergence avoids the instability in the exponential function, its $(f')^{-1}(x)$ value is negative when $x < -2$.

To take advantage of both divergences while avoiding their drawbacks, we propose the following *piecewise* $f$-divergence that has properties similar to Pearson $\chi^2$ when $x$ is large while has properties similar to Reverse KL when $x$ is small:

$$f(x) = \begin{cases} (x-1)^2 & x \geq 1, \\ x \log x - x + 1 & 0 \leq x < 1. \end{cases} \tag{8}$$

In Appendix B, we prove that $f(x)$ is a well-defined $f$-divergence. We also prove that $f(x)$ has the following proposition:

**Proposition 1.** *Given the piecewise $f$-divergence in Eq.(8), $(f')^{-1}(x)$ and $f^*(x)$ has the following formulation:*

$$(f')^{-1}(x) = \begin{cases} \frac{x}{2} + 1 & x \geq 0 \\ e^x & x < 0 \end{cases} \qquad f^*(x) = \begin{cases} \frac{x^2}{4} + x & x \geq 0 \\ e^x - 1 & x < 0 \end{cases} \tag{9}$$

We can see that given any $x$, the exponential function in $(f')^{-1}(x)$ ensures a strictly positive value. Meanwhile, $f^*(x)$ is at most a quadratic polynomial, which ensures numerical stability in practice. Note that $f(x)$ enables a stable policy transformation and can also be applied to other DICE-based methods when the optimal stationary distribution ratio needs to be positive.

### 3.3   Boost Performance with Optimal Action Selection

After transforming $\pi^{\mathcal{D}}$ into in-support $\pi^*$, we now introduce the select-step to boost the performance during evaluation. One issue is the multi-modality in the optimal policy, this partially arises from the policy constraint used to regularize its output [9, 25, 55, 34]. Under policy constraint, the regularized optimal policy is unlikely to be deterministic and may be multi-modal.

More specifically, in our method, as we strictly follow the score function $\nabla_{a_t} \log \pi_t^*(a_t|s)$ of in-support optimal policy $\pi^*$ to generate high-value actions during the reverse diffusion process, any mode that has non-zero probability under $\pi^*(a|s)$ could be generated, which lead to a sub-optimal choice. Similar issues have been discussed in previous works [14, 13]. In Diffusion-DICE, it's natural to leverage the optimal critic $Q^*$ derived from DICE in Eq.(3) to identify the best action. Given a specific state $s$, we first follow $\nabla_{a_t} \log \pi_t^*(a_t|s)$ in the reverse diffusion process to generate a few actions from $\pi^*(a|s)$. Then we evaluate these actions with $Q^*$ and select the action with the highest $Q^*(s,a)$ value as the policy's output as

$$\pi(s) = a^* \triangleq \underset{a \in \{a_0^1, ..., a_0^K \sim \pi^*(a|s)\}}{\arg\max} Q^*(s,a). \tag{10}$$

Different from previous guide-only or select-only methods, we base the select stage on the guide stage. As mentioned before, using IGL accurately guides the generated candidate actions towards in-support actions with high value. With the assistance of the guide stage, we can sample candidates from $\pi^*(a|s)$, which has a high probability of being high-quality. This means only a small number of candidates are required to be sampled, which reduces the probability of sampling OOD actions. This leads to minimal error exploitation while still attaining high returns. We give an illustration of the guide-then-select paradigm in Figure 1.

---
**Algorithm 1** Diffusion-DICE

1: Initialize value function $Q_{\phi_1}$, $V_{\phi_2}$, diffusion behavior model $\epsilon_\psi$, guidance network $g_\theta$
2: // Training
3: Pretrain the diffusion behavior model by minimizing $\mathbb{E}_{\mathcal{U}(t),(s,a)\sim\mathcal{D},\epsilon\sim\mathcal{N}(\mathbf{0},\mathbf{I})}[\|\epsilon_\psi(\alpha_t a + \sigma_t \epsilon, s, t) - \epsilon\|^2]$
4: **for** $t = 1, 2, \cdots, N$ **do**
5:     Sample transitions $(s, a, r, s') \sim \mathcal{D}$
6:     Update $Q_{\phi_1}$ and $V_{\phi_2}$ by minimizing Eq.(3)
7:     Update $g_\theta$ by minimizing Eq.(7)
8: **end for**
9: // Evaluation
10: Sample $\{a_0^i\}_{i=1}^K$ from $\pi^*$ following Eq.(5), with $\nabla_{a_t} \log \pi_t^*(a_t|s) = \frac{\epsilon_\psi(a_t,s,t)}{-\sigma_t} + \nabla_{a_t} g_\theta(s, a_t, t)$
11: Select action according to Eq.(10)

---

We term this *guide-then-select* method Diffusion-DICE and present its pseudo-code in Algorithm 1. During the training stage, Diffusion-DICE estimates the optimal stationary distribution ratio using DICE with our piecewise $f$-divergence and calculates the guidance with IGL, both in an in-sample manner. During the testing stage,

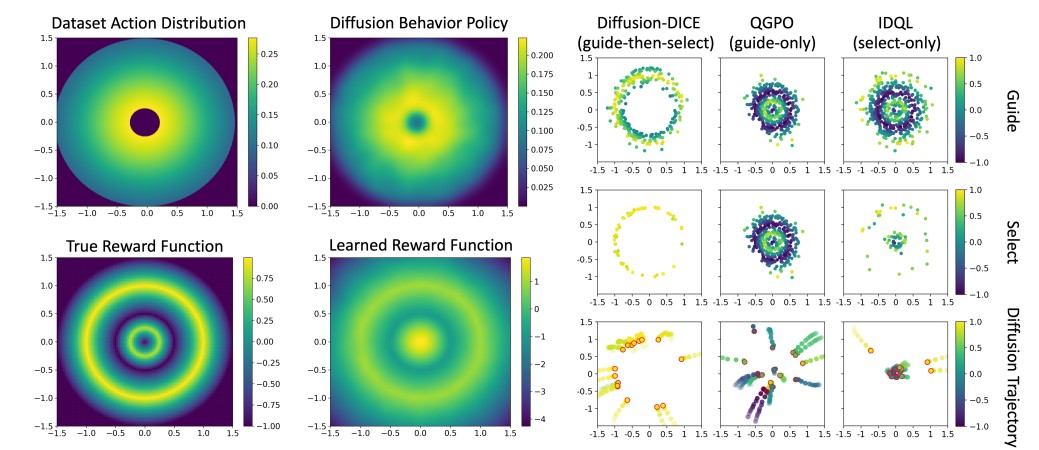

Figure 2: Toycase of a 2-D bandit problem. The action in the offline dataset follows a bivariate standard normal distribution constrained within an annular region. The ground truth reward has two peaks extending from the center outward. We use a diffusion model $\hat{\pi}^{\mathcal{D}}$ to fit the behavior policy and a reward model $\hat{R}$ to fit the ground truth reward $R$. Both $\hat{\pi}^{\mathcal{D}}$ and $\hat{R}$ fit in-distribution data well while making error in out-of-distribution regions. Diffusion-DICE could generate correct optimal actions in the outer circle while other methods tend to expolit error information from $\hat{R}$ and only generate overestimated, sub-optimal actions.

Diffusion-DICE selects the action from $\pi^*(a|s)$ with the highest value. By selecting from a small number of action candidates from $\pi^*(a|s)$, minimal error of the action evaluation model will be exploited. Besides serving as a diffusion-based offline RL algorithm, Diffusion-DICE also introduces a novel policy extraction method from other DICE-based algorithms. By leveraging expressive generative models, Diffusion-DICE can capture the multi-modality in $\pi^*$ and better draw upon the knowledge from $\frac{d^*(s,a)}{d^{\mathcal{D}}(s,a)}$ and $\pi^{\mathcal{D}}$.

**Toycase validation.** We use a toycase to validate that the *guide-then-select* paradigm used in Diffusion-DICE indeed brings minimal error exploitation that other diffusion-based methods suffer from. Here we aim to solve a 2-D bandit problem given a fixed action dataset. The action space is continuous and actions in the offline dataset follow a bivariate standard normal distribution constrained within an annular region. We show the dataset distribution $\pi^{\mathcal{D}}$, the learned diffusion behavior policy $\hat{\pi}^{\mathcal{D}}$, the ground truth reward $R$ and the predicted reward $\hat{R}$ in Figure 2. Note that the true optimal reward occurs on the outer circle. However, due to limited data coverage, the learned reward function exploits overestimation error on unseen regions, e.g., actions inside the inner circle have erroneous high values. What's worse, due to fitting errors, the diffusion behavior policy may generate such overestimated actions. We take Diffusion-DICE with a guide-based method, QGPO [32], and a select-based method, IDQL [14] for comparison.

More specifically, we visualize the sample generation process in different methods. For guide-based method QGPO, as it utilizes actions generated by $\hat{\pi}^{\mathcal{D}}$ and the value of $\hat{R}$ on them to obtain guidance, the guidance exploits overestimation errors of OOD actions in the center and drifts the generated actions towards them. Note that adding a select step in QGPO will not help as the guide-step is already incorrect. IDQL, because it is free of guidance, requires a large number of generated candidate actions to ensure the coverage of optimal action. IDQL utilizes $\hat{R}$ to select from those suboptimal or out-of-distribution actions, which however, are the sources of error exploitation in $\hat{R}$. For Diffusion-DICE, because the first guide-step accurately guides the generated actions towards in-support while high-value regions (i.e., actions around the outer circle), in the select step, the learned value function $\hat{R}$ could make a correct evaluation and successfully select the true best actions in the outer circle.

Table 1: Evaluation results on D4RL benchmark. We report the average normalized scores at the end of training with standard deviation across 5 random seeds. Diffusion-DICE (D-DICE) demonstrates superior performance compared to all baseline algorithms in 13 out of 15 tasks, especially on more challenging tasks.

| D4RL Dataset | Gaussian policy | | | | Diffusion policy | | | | |
|---|---|---|---|---|---|---|---|---|---|
| | CQL | IQL | SQL | O-DICE | Diffusion-QL | SfBC | QGPO | IDQL | D-DICE (ours) |
| halfcheetah-m | 44.0 | 47.4 | 48.3 | 47.4 | 51.1 ±0.5 | 45.9 ±2.2 | 54.1 ±0.4 | 51.0 | 60.0 ±0.6 |
| hopper-m | 58.5 | 66.3 | 75.5 | 86.1 | 90.5 ±4.6 | 57.1 ±4.1 | 98.0±2.6 | 65.4 | 100.2 ±3.2 |
| walker2d-m | 72.5 | 72.5 | 84.2 | 84.9 | 87.0 ±0.9 | 77.9 ±2.5 | 86.0±0.7 | 82.5 | 89.3 ±1.3 |
| halfcheetah-m-r | 45.5 | 44.2 | 44.8 | 44.0 | 47.8 ±0.3 | 37.1 ±1.7 | 47.6±1.4 | 45.9 | 49.2 ±0.9 |
| hopper-m-r | 95.0 | 95.2 | 99.7 | 99.9 | 101.3 ±0.6 | 86.2 ±9.1 | 96.9±2.6 | 92.1 | 102.3 ±2.1 |
| walker2d-m-r | 77.2 | 76.1 | 81.2 | 83.6 | 95.5 ±1.5 | 65.1 ±5.6 | 84.4±4.1 | 85.1 | 90.8±2.6 |
| halfcheetah-m-e | 90.7 | 86.7 | 94.0 | 93.2 | 96.8 ±0.3 | 92.6 ±0.5 | 93.5±0.3 | 95.9 | 97.3 ±0.6 |
| hopper-m-e | 105.4 | 101.5 | 111.8 | 110.8 | 111.1 ±1.3 | 108.6 ±2.1 | 108.0±2.5 | 108.6 | 112.2 ±0.3 |
| walker2d-m-e | 109.6 | 110.6 | 110.0 | 110.8 | 110.1 ±0.3 | 109.8 ±0.2 | 110.7±0.6 | 112.7 | 114.1 ±0.5 |
| antmaze-u | 84.8 | 85.5 | 92.2 | 94.1 | 93.4 ±3.4 | 92.0 ±2.1 | 96.4±1.4 | 94.0 | 98.1 ±1.8 |
| antmaze-u-d | 43.4 | 66.7 | 74.0 | 79.5 | 66.2 ±8.6 | 85.3 ±3.6 | 74.4±9.7 | 80.2 | 82.0±8.4 |
| antmaze-m-p | 65.2 | 72.2 | 80.2 | 86.0 | 76.6 ±10.8 | 81.3 ±2.6 | 83.6±4.4 | 84.5 | 91.3 ±3.1 |
| antmaze-m-d | 54.0 | 71.0 | 79.1 | 82.7 | 78.6 ±10.3 | 82.0 ±3.1 | 83.8±3.5 | 84.8 | 85.7 ±4.8 |
| antmaze-l-p | 38.4 | 39.6 | 53.2 | 55.9 | 46.4 ±8.3 | 59.3 ±14.3 | 66.6±9.8 | 63.5 | 68.6 ±8.6 |
| antmaze-l-d | 31.6 | 47.5 | 52.3 | 54.0 | 56.6 ±7.6 | 45.5 ±6.6 | 64.8±5.5 | 67.9 | 72.0 ±6.5 |
| kitchen-p | 49.8 | 46.3 | - | - | 60.5±6.9 | - | - | - | 78.3 ±3.4 |
| kitchen-m | 51.0 | 51.0 | - | - | 62.6 ±5.1 | - | - | - | 67.8 ±4.2 |
| pen-human | 37.5 | 71.5 | - | - | 72.8 ±2.7 | - | 73.9±4.6 | - | 84.4 ±2.7 |
| pen-cloned | 39.2 | 37.3 | - | - | 57.3±3.6 | - | 54.2 ±5.1 | - | 83.8 ±3.2 |

# 4 Experiments

In this section, we present empirical evaluations of Diffusion-DICE. To validate Diffusion-DICE's ability to transform the behavior policy into the optimal policy while maintaining minimal error exploitation, we conduct experiments on two fronts. On one hand, we evaluate Diffusion-DICE on the D4RL offline RL benchmark and compare it with other strong diffusion-based and DICE-based methods. On the other hand, we use alternative criteria to demonstrate that the *guide-then-select* paradigm results in minimal error exploitation. Experimental details are shown in Appendix D.

## 4.1 D4RL Benchmark Datasets:

We first evaluate Diffusion-DICE on the D4RL benchmark [8] and compare it with several related algorithms. For the evaluation tasks, we select MuJoCo locomotion tasks and AntMaze navigation tasks. While MuJoCo locomotion tasks are popular in offline RL, AntMaze navigation tasks are more challenging due to their stronger need for trajectory stitching. For baseline algorithms, we selected state-of-the-art methods not only from traditional methods that use Gaussian-policy (including DICE-based methods) but also from diffusion-based methods. Gaussian-policy-based baseline includes CQL [26], in-sample based methods IQL [24], SQL [55] and DICE-based method O-DICE [34]. Notably, O-DICE is a recently proposed DICE-based algorithm that stands out among various DICE-based offline RL methods. Diffusion-policy-based baseline includes Diffusion-QL [49] SfBC [4], QGPO [32] and IDQL [14]. We also compare Diffusion-DICE with its Gaussian-policy counterpart to show the benefit of using Diffusion-policy in Appendix D. While SfBC and IDQL simply sample from behavior policy candidates and select according to the action evaluation model, Diffusion-QL and QGPO will guide generated actions towards high-value ones. It is worth noting that Diffusion-QL will also resample from generated actions after the guidance.

The results show that Diffusion-DICE outperforms all other baseline algorithms, including previous SOTA diffusion-based methods, especially on MuJoCo medium, medium-replay datasets, and AntMaze datasets. The consistently better performance compared with Diffusion-QL and QGPO demonstrates the essentiality of in-sample guidance learning. Compared with SfBC and IDQL, Diffusion-DICE also shows superior performance, even with fewer action candidates. This is because the guide stage provides the in-support optimal policy for the select stage to sample from, which underscores the necessity of sampling carefully from an in-support optimal action distribution, rather than from the behavior distribution. We refer to the comparison of candidate numbers under different environments in Appendix D. Furthermore, the substantial performance gap between Diffusion-DICE and

O-DICE reflects the multi-modality of DICE's optimal policy and firmly positions Diffusion-DICE as a superior policy extraction method for other DICE-based algorithms.

## 4.2 Further Experiments on Error Exploitation

We then continue to verify that the guide-then-select paradigm used in Diffusion-DICE indeed exploits minimal error. This is obvious for the guide stage because we only leverage in-sample actions for both critic training and guidance learning. For the select stage, additional evidence is needed, as Diffusion-DICE also selects actions from generated candidates. Our key observation is that if the action value remains high during evaluation but results in a low return trajectory, these action values must suffer from overestimation error.

Based on this, we choose to compare the average state-action value and the average return between actions selected by our guide-then-select paradigm and simply select from the behavior policy. Given identical critic and behavior policy, we run Diffusion-DICE and its guidance-free variant for 10 episodes, comparing their average $Q(s,a)$ values and their normalized scores. The results are shown in Figure 3. We also compare the learning curves of their average $Q(s,a)$ values over time in Appendix D.

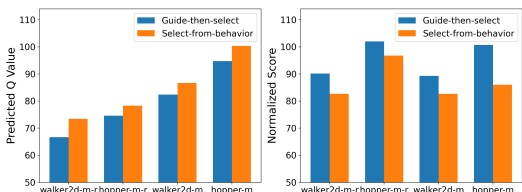

Figure 3: Actions generated by the guide-then-select paradigm result in better performance while have less overestimation error.

It's clear from the results that our guide-then-select paradigm achieves better performance while having less overestimated $Q(s,a)$ values. This result validates the minimal error exploitation in Diffusion-DICE.

## 5 Related Work

**Offline RL** To tackle the distributional shift problem, most model-free offline RL methods augment existing off-policy RL methods with a behavior regularization term. Behavior regularization can appear explicitly as divergence penalties [50, 25, 51, 9], implicitly through weighted behavior cloning [48, 38, 52], or more directly through careful parameterization of the policy [11, 58]. Another way to apply action-level regularization is via modification of value learning objective to incorporate some form of regularization, to encourage staying near the behavioral distribution and being pessimistic about OOD state-action pairs [26, 24, 55, 47]. There are also several works incorporating action-level regularization through the use of uncertainty [2] or distance function [29].

All these methods are based on the actor-critic framework and use unimodal Gaussian policy. However, several works indicate their limited ability to model multi-modal policy distribution [49, 14, 4], thereby leading to suboptimal performance. To remedy this, it's natural to employ powerful generative models to represent the policy. DT [5] uses transformer as the policy, Diffusion-QL [49], SfBC [4], QGPO [32] and IDQL [14] leverage diffusion models to represent policy. Other generative models like CVAE and consistency model have also been used as policy in offline RL [58, 54, 7]. Another line of methods utilizes diffusion models for trajectory-level planning by generating high-return trajectories and taking the corresponding action of the current state [17, 1]. While generative models are widely used in offline RL, few methods consider the existing errors in these models and whether the generation process exploits these errors.

**DICE-based methods** The core of DICE-based methods revolves around the ratio of the stationary distribution between two policies. This ratio can serve as the estimation target in off-policy evaluation [36, 57] or as part of a state-action level constraint term in RL [37, 41, 34]. In offline IL [53], it can connect the optimal stationary distribution of a regularized MDP with the dataset distribution [22, 12, 19]. In constrained RL, this ratio can induce the discounted sum of cost without an additional function approximator [28]. Under the imperfect rewards setting, this ratio can reflect the gap between the given rewards and the underlying perfect rewards [30]. All of these DICE methods consider this ratio as a transformation from one stationary distribution to another. In this paper, however, we extend this ratio as a transformation from the behavior policy to the optimal policy and propose a novel way of doing so by using diffusion models, which also serve as a replacement for the Gaussian-based policy extraction in DICE.

## 6 Conclusion

In this work, we propose a new diffusion-based offline RL algorithm, Diffusion-DICE. Diffusion-DICE uses a guide-then-select paradigm to select the best in-support actions while achieving minimal error exploitation in the value function. Diffusion-DICE also serves as a replacement for the Gaussian-based policy extraction part in current DICE methods, successfully unleashing the power of DICE-based methods. Through toycase illustration and extensive experiments, we show that Diffusion-DICE outperforms prior SOTA methods on a variety of datasets, especially those with multi-modal complex behavior distribution. One limitation of Diffusion-DICE is the sampling process of diffusion models is costly and slow. Another limitation is the training of diffusion models may suffer in low-data regimes. One future work is using more advanced generative models [44, 40] as a better choice.

## Acknowledgement

HX thanks Harshit Sikchi for insightful discussions. The SJTU team is partially supported by Shanghai Municipal Science and Technology Major Project (2021SHZDZX0102) and National Natural Science Foundation of China (62322603, 62076161). HX and AZ are supported by NSF 2340651.

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

# A  A More Detailed Discussion of DICE and Diffusion Model in RL

**Derivation of learning objectives in DICE**     DICE algorithms consider the following regularized RL problem as a convex programming problems with Bellman-flow constraints and apply Fenchel-Rockfeller duality or Lagrangian duality to solve it. The regularization term aims at imposing state-action level constraints.[35, 27, 34].

$$\max_{\pi} \mathbb{E}_{(s,a)\sim d^{\pi}}[r(s,a)] - \alpha D_f(d^{\pi}(s,a)\|d^{\mathcal{D}}(s,a)) \tag{11}$$

Here $D_f(d^{\pi}(s,a)\|d^{\mathcal{D}}(s,a))$ is the $f$-divergence which is defined with $D_f(P\|Q) = \mathbb{E}_{\omega\in Q}\left[f\left(\frac{P(\omega)}{Q(\omega)}\right)\right]$. Directly solving $\pi^*$ is impossible because it's intractable to calculate $d^{\pi}(s,a)$. However, one can change the optimization variable from $\pi$ to $d^{\pi}$ because of the bijection existing between them. Then with the assistance of Bellman-flow constraints, we can obtain an optimization problem with respect to $d$:

$$\max_{d\geq 0}\mathbb{E}_{(s,a)\sim d}[r(s,a)] - \alpha D_f(d(s,a)\|d^{\mathcal{D}}(s,a))$$

$$\text{s.t. } \sum_{a\in\mathcal{A}} d(s,a) = (1-\gamma)d_0(s) + \gamma \sum_{(s',a')} d(s',a')p(s|s',a'), \forall s\in\mathcal{S} \tag{12}$$

Note that the feasible region has to be $\{d : \forall s\in\mathcal{S}, a\in\mathcal{A}, d(s,a)\geq 0\}$ because $d$ should be non-negative to ensure a valid corresponding policy. After applying Lagrangian duality, we can get the following optimization target following [27]:

$$\min_{V(s)}\max_{d\geq 0}\mathbb{E}_{(s,a)\sim d}[r(s,a)] - \alpha D_f(d(s,a)\|d^{\mathcal{D}}(s,a))$$

$$+ \sum_{s} V(s)\Big((1-\gamma)d_0(s) + \gamma\sum_{(s',a')}d(s',a')p(s|s',a') - \sum_{a}d(s,a)\Big)$$

$$= \min_{V(s)}\max_{\omega\geq 0}(1-\gamma)\mathbb{E}_{d_0(s)}[V(s)]$$

$$+ \mathbb{E}_{s,a\sim d^{\mathcal{D}}}\Big[\omega(s,a)\Big(r(s,a) + \gamma\sum_{s'}p(s'|s,a)V(s') - V(s)\Big)\Big] - \alpha\mathbb{E}_{s,a\sim d^{\mathcal{D}}}\Big[f(\omega(s,a))\Big] \tag{13}$$

Here we denote $\omega(s,a)$ as $\frac{d(s,a)}{d^{\mathcal{D}}(s,a)}$ for simplicity. By incorporating the non-negative constraint of $d$ and again solving the constraint problem with Lagrangian duality, we can derive the optimal solution $w^*(s,a)$ for the inner problem and thus reduce the bilevel optimization problem to the following optimization problem:

$$\min_{V(s)}(1-\gamma)\mathbb{E}_{d_0(s)}[V(s)] + \alpha\mathbb{E}_{s,a\sim d^{\mathcal{D}}}\Big[f^*\big(\frac{r(s,a) + \gamma\sum_{s'}p(s'|s,a)V(s') - V(s)}{\alpha}\big)\Big] \tag{14}$$

Here $f^*$ is a variant of $f$'s convex conjugate. Note that in the offline RL setting, the inner summation $\sum_{s'}p(s'|s,a)V(s')$ is usually intractable because of limited data samples. To handle this issue and increase training stability, DICE methods usually use additional network $Q(s,a)$ to fit $r(s,a) + \gamma\sum_{s'}p(s'|s,a)V(s')$, by optimizing the following MSE objective:

$$\min_{Q}\mathbb{E}_{(s,a,s')\sim d^{\mathcal{D}}}\Big[\big(r(s,a) + \gamma V(s') - Q(s,a)\big)^2\Big] \tag{15}$$

And because of doing so, the optimization objective in 14 can be replaced with:

$$\min_{V}\mathbb{E}_{s\sim d^0}[(1-\gamma)V(s)] + \mathbb{E}_{(s,a)\sim d^{\mathcal{D}}}\big[\alpha f^*\big([Q(s,a) - V(s)]/\alpha\big)\big] \tag{16}$$

Also note that to increase the diversity of samples, one often extends the distribution of initial state $d_0$ to $d^{\mathcal{D}}$ by treating every state in a trajectory as initial state [22].

**Discussion of diffusion policy and other methods in offline RL**     In offline RL, as discussed before, most diffusion-based algorithms utilize diffusion models to represent policy $\pi$. This policy can either be the behavior policy [14, 4], the optimal policy [49, 32]. When representing policy with diffusion model, the ultimate goal is to model the optimal policy $\pi^*$ with diffusion model. Different

from other tasks in image generation or video generation [43, 16], RL tasks will not provide data samples under $\pi^*$, while the agent has to learn such optimal policy. This makes directly leveraging the diffusion model to fit the optimal policy impossible. However, besides these methods that are based on diffusion policy, some methods utilize a diffusion model to fit the entire trajectory's distribution [17, 1]. These methods try to generate the optimal trajectory based on the current state. After generating the entire trajectory, these methods simply take the action corresponding to the current state. It's worth noting that although these methods avoid modeling $\pi^*$, extra effort is required to generate high-quality trajectories.

## B  Additional Proofs

**Optimal policy transformation with stationary distribution ratio**: Given that $\pi^{\mathcal{D}}$ is the behavior policy, $\frac{d^*(s,a)}{d^{\mathcal{D}}(s,a)}$ and $\pi^*$ are the optimal stationary distribution correction and its corresponding optimal policy. $\pi^*$ and $\pi^{\mathcal{D}}$ satisfy the following proposition:

$$\pi^*(a|s) = \frac{\frac{d^*(s,a)}{d^{\mathcal{D}}(s,a)}}{\int_{a\in\mathcal{A}}\frac{d^*(s,a)}{d^{\mathcal{D}}(s,a)}\pi^{\mathcal{D}}(a|s)da}\pi^{\mathcal{D}}(a|s) \tag{17}$$

*Proof.* Using the definition between $d^{\pi}(s)$ and $d^{\pi}(s,a)$, we have for any $\pi$, $d^{\pi}(s,a) = d^{\pi}(s)\cdot\pi(a|s)$. Then we have:

$$\begin{aligned}
\frac{\frac{d^*(s,a)}{d^{\mathcal{D}}(s,a)}}{\int_{a\in\mathcal{A}}\frac{d^*(s,a)}{d^{\mathcal{D}}(s,a)}\pi^{\mathcal{D}}(a|s)da}\pi^{\mathcal{D}}(a|s) &= \frac{\frac{d^*(s,a)}{d^{\mathcal{D}}(s,a)}}{\frac{\int_{a\in\mathcal{A}}d^*(s,a)da}{d^{\mathcal{D}}(s)}}\pi^{\mathcal{D}}(a|s) \\
&= \frac{d^*(s,a)}{d^{\mathcal{D}}(s,a)}\cdot\frac{d^{\mathcal{D}}(s)}{d^*(s)}\pi^{\mathcal{D}}(a|s) \\
&= \pi^*(a|s)
\end{aligned} \tag{18}$$

$\square$

**Verification of Eq.(6)**: The score function of the optimal policy $\pi^*(a|s)$ in DICE and the behavior policy $\pi^{\mathcal{D}}(a|s)$ satisfy: $\nabla_{a_t}\log\pi_t^*(a_t|s) = \nabla_{a_t}\log\pi_t^{\mathcal{D}}(a_t|s) + \nabla_{a_t}\log\mathbb{E}_{a_0\sim\pi^{\mathcal{D}}(a_0|a_t,s)}[\frac{d^*(s,a_0)}{d^{\mathcal{D}}(s,a_0)}]$.

We mainly follow the proof in Lu et al. [32] to verify this equation. Given the relationship between $\pi^*(a|s)$ and $\pi^*(a|s)$ in Eq.(18) we have:

$$\begin{aligned}
\pi_t^*(a_t|s) &= \int p(a_t|a_0)\pi_0^*(a_0|s)da_0 \\
&= \int p(a_t|a_0,s)\pi^*(a_0|s)da_0 \\
&= \int p(a_t|a_0,s)\frac{\frac{d^*(s,a_0)}{d^{\mathcal{D}}(s,a_0)}}{\int_{a\in\mathcal{A}}\frac{d^*(s,a)}{d^{\mathcal{D}}(s,a)}\pi^{\mathcal{D}}(a|s)da}\pi^{\mathcal{D}}(a_0|s)da_0 \\
&= \int p(a_t|a_0,s)\pi_0^{\mathcal{D}}(a_0|s)\frac{\frac{d^*(s,a_0)}{d^{\mathcal{D}}(s,a_0)}}{\int_{a\in\mathcal{A}}\frac{d^*(s,a)}{d^{\mathcal{D}}(s,a)}\pi^{\mathcal{D}}(a|s)da}da_0 \\
&= \int \pi^{\mathcal{D}}(a_0|a_t,s)\pi_t^{\mathcal{D}}(a_t|s)\frac{\frac{d^*(s,a_0)}{d^{\mathcal{D}}(s,a_0)}}{\int_{a\in\mathcal{A}}\frac{d^*(s,a)}{d^{\mathcal{D}}(s,a)}\pi^{\mathcal{D}}(a|s)da}da_0 \\
&= \pi_t^{\mathcal{D}}(a_t|s)\mathbb{E}_{\pi^{\mathcal{D}}(a_0|a_t,s)}\left[\frac{\frac{d^*(s,a_0)}{d^{\mathcal{D}}(s,a_0)}}{\int_{a\in\mathcal{A}}\frac{d^*(s,a)}{d^{\mathcal{D}}(s,a)}\pi^{\mathcal{D}}(a|s)da}\right]
\end{aligned} \tag{19}$$

Note that the second equation holds because $p(a_t|a_0)$ represents the forward transition probability in the diffusion process, which is unrelated to $s$ and $\pi_0^*$ is exactly the optimal policy $\pi^*$. The

forth equation holds for the similar reasons. We exchange the condition variable to derive the final relationship between $\pi_t^*(a|s)$ and $\pi_t^*(a|s)$. Having this relationship, we can directly calculate their score functions as follows:

$$\nabla_{a_t} \log \pi_t^*(a_t|s) = \nabla_{a_t} \log \pi_t^{\mathcal{D}}(a_t|s) + \nabla_{a_t} \log \mathbb{E}_{a_0 \sim \pi^{\mathcal{D}}(a_0|a_t,s)}\left[\frac{\frac{d^*(s,a_0)}{d^{\mathcal{D}}(s,a_0)}}{\int_{a \in \mathcal{A}} \frac{d^*(s,a)}{d^{\mathcal{D}}(s,a)} \pi^{\mathcal{D}}(a|s)da}\right]$$

$$= \nabla_{a_t} \log \pi_t^{\mathcal{D}}(a_t|s) + \nabla_{a_t} \log \frac{\mathbb{E}_{a_0 \sim \pi^{\mathcal{D}}(a_0|a_t,s)}\left[\frac{d^*(s,a_0)}{d^{\mathcal{D}}(s,a_0)}\right]}{\int_{a \in \mathcal{A}} \frac{d^*(s,a)}{d^{\mathcal{D}}(s,a)} \pi^{\mathcal{D}}(a|s)da} \tag{20}$$

$$= \nabla_{a_t} \log \pi_t^{\mathcal{D}}(a_t|s) + \nabla_{a_t} \log \mathbb{E}_{a_0 \sim \pi^{\mathcal{D}}(a_0|a_t,s)}\left[\frac{d^*(s,a_0)}{d^{\mathcal{D}}(s,a_0)}\right]$$

Note that the second equation holds because the integral $\int_{a \in \mathcal{A}} \frac{d^*(s,a)}{d^{\mathcal{D}}(s,a)} \pi^{\mathcal{D}}(a|s)da$ is unrelated to $a_0$, $a_t$ and because of so, its gradient with respect to $a_t$ becomes 0.

**Lemma 1.** *Given a random variable $X$ and its corresponding distribution $P(X)$, for any non-negative function $f(x)$, the following problem is convex and its optimizer is given by $y^* = \log \mathbb{E}_{x \sim P(X)}[f(x)]$.*

$$\min_y \mathbb{E}_{x \sim P(X)}[f(x) \cdot e^{-y} + y] \tag{21}$$

*Proof.* Because $y$ is the constant optimizer, this problem can be reformulated as:

$$\min_y \mathbb{E}_{x \sim P(X)}[f(x)] \cdot e^{-y} + y \tag{22}$$

We first verify that this problem is actually a convex problem with respect to $y$. Because $f(x)$ is a non-negative function, we can take the second derivative of this objective and get:

$$\mathbb{E}_{x \sim P(X)}[f(x)] \cdot e^{-y} \tag{23}$$

Because $f(x)$ is a non-negative function, the second derivative is also non-negative, which means this problem is a convex problem with respect to $y$. For a convex problem, the optimizer can be induced from the first-order condition:

$$\nabla_{y^*}(\mathbb{E}_{x \sim P(X)}[f(x)] \cdot e^{-y^*} + y^*) = 0$$
$$y^* = \log \mathbb{E}_{x \sim P(X)}[f(x)] \tag{24}$$

$\square$

**Theorem 1.** $\nabla_{a_t} \log \mathbb{E}_{\pi^{\mathcal{D}}(a_0|a_t,s)}[w(s,a_0)]$ *can be obtained by solving the following optimization problem:*

$$\min_\theta \mathbb{E}_{t \sim \mathcal{U}(0,T)} \mathbb{E}_{\pi^{\mathcal{D}}(a_0|s)} \mathbb{E}_{p(a_t|a_0)}\left[w(s,a_0)e^{-g_\theta(s,a_t,t)} + g_\theta(s,a_t,t)\right] \tag{25}$$

*The optimal solution $\theta^*$ satisfies $\nabla_{a_t} g_{\theta^*}(s,a_t,t) = \nabla_{a_t} \log \mathbb{E}_{\pi^{\mathcal{D}}(a_0|a_t,s)}[w(s,a_0)]$.*

*Proof.* Because the forward diffusion process simply adds noise to $a_0$ and is unrelated to $s$, taking $s$ as a condition doesn't affect $p(a_t|a_0)$. This means $p(a_t|a_0,t) = p(a_t|a_0)$. By changing the conditional variables from $(a_0)^{(1:K)}$ to $(a_t)^{(1:K)}$ we have:

$$\mathbb{E}_{t \sim \mathcal{U}(0,T)} \mathbb{E}_{\pi^{\mathcal{D}}(a_0|s)} \mathbb{E}_{p(a_t|a_0)}\left[w(s,a_0)e^{-g_\theta(s,a_t,t)} + g_\theta(s,a_t,t)\right]$$

$$= \mathbb{E}_{t \sim \mathcal{U}(0,T)} \mathbb{E}_{\pi^{\mathcal{D}}(a_0|s)} \mathbb{E}_{p(a_t|a_0,s)}\left[w(s,a_0)e^{-g_\theta(s,a_t,t)} + g_\theta(s,a_t,t)\right]$$

$$= \mathbb{E}_{t \sim \mathcal{U}(0,T)} \mathbb{E}_{\pi^{\mathcal{D}}(a_t,a_0|s)}\left[w(s,a_0)e^{-g_\theta(s,a_t,t)} + g_\theta(s,a_t,t)\right] \tag{26}$$

$$= \mathbb{E}_{t \sim \mathcal{U}(0,T)} \mathbb{E}_{\pi^{\mathcal{D}}(a_t|s)} \mathbb{E}_{\pi^{\mathcal{D}}(a_0|a_t,s)}\left[w(s,a_0)e^{-g_\theta(s,a_t,t)} + g_\theta(s,a_t,t)\right]$$

$$= \mathbb{E}_{t \sim \mathcal{U}(0,T)} \mathbb{E}_{\pi^{\mathcal{D}}(a_t|s)}\left[\mathbb{E}_{\pi^{\mathcal{D}}(a_0|a_t,s)}[w(s,a_0)]e^{-g_\theta(s,a_t,t)} + g_\theta(s,a_t,t)\right]$$

We first take the first-order derivative with respect to each $g_\theta(s, a_t, t)$ for this objective and solve its saddle point:

$$\nabla_{g_\theta(s,a_t,t)} \mathbb{E}_{t\sim\mathcal{U}(0,T)} \mathbb{E}_{\pi^{\mathcal{D}}(a_t|s)} \left[ \mathbb{E}_{\pi^{\mathcal{D}}(a_0|a_t,s)} [w(s,a_0)] e^{-g_\theta(s,a_t,t)} + g_\theta(s,a_t,t) \right]$$
$$= \frac{1}{T} \cdot \pi^{\mathcal{D}}(a_t|s) \cdot \left( -\mathbb{E}_{\pi^{\mathcal{D}}(a_0|a_t,s)} [w(s,a_0)] e^{-g_\theta(s,a_t,t)} + 1 \right) \tag{27}$$

Because any possible intermediate action $a_t$ has a positive value of $\pi^{\mathcal{D}}(a_t|s)$, the saddle point $g_{\theta*}(s, a_t, t)$ must satisfy:

$$-\mathbb{E}_{\pi^{\mathcal{D}}(a_0|a_t,s)} [w(s,a_0)] e^{-g_{\theta*}(s,a_t,t)} + 1 = 0$$
$$g_{\theta*}(s, a_t, t) = \log \mathbb{E}_{\pi^{\mathcal{D}}(a_0|a_t,s)} [w(s,a_0)] \tag{28}$$

and then $\nabla_{a_t} g_{\theta*}(s, a_t, t) = \nabla_{a_t} \log \mathbb{E}_{\pi^{\mathcal{D}}(a_0|a_t,s)} [w(s,a_0)]$

We then take the second-order derivative with respect to each $g_\theta(s, a_t, t)$ for this objective:

$$\nabla^2_{g_\theta(s,a_t,t)} \mathbb{E}_{t\sim\mathcal{U}(0,T)} \mathbb{E}_{\pi^{\mathcal{D}}(a_t|s)} \left[ \mathbb{E}_{\pi^{\mathcal{D}}(a_0|a_t,s)} [w(s,a_0)] e^{-g_\theta(s,a_t,t)} + g_\theta(s,a_t,t) \right]$$
$$= \frac{1}{T} \cdot \pi^{\mathcal{D}}(a_t|s) \cdot \mathbb{E}_{\pi^{\mathcal{D}}(a_0|a_t,s)} [w(s,a_0)] e^{-g_\theta(s,a_t,t)} \tag{29}$$

Because $w(s, a_0)$ has to be non-negative for any $a_0$ to ensure a valid transformation from $\pi^{\mathcal{D}}$ to $\pi^*$, it's obvious that the second-order derivative is always non-negative. This means this objective is convex with respect to $g_\theta(s, a_t, t)$ and has convergence guarantee given unlimited model capacity. $\qquad\square$

**Lemma 2.** *The following $f_p(x)$ is a well-defined $f$-divergence:*

$$f_p(x) = \begin{cases} (x-1)^2 & x \geq 1 \\ x\log x - x + 1 & 0 \leq x < 1 \end{cases} \tag{30}$$

*Proof.* By the definition of $f$-divergence in Rényi [39], $f_p(x)$ should be a convex function from $[0, +\infty) \rightarrow (-\infty, +\infty]$. $f_p(x)$ should be finite for all $x > 0$ and $f_p(1) = 0$, $f_p(0) = \lim_{x\to 0+} f_p(x)$.

We first verify that $f_p(x)$ is convex. By taking first-order and second-order derivatives on $f_p(x)$ we have:

$$f_p'(x) = \begin{cases} 2(x-1) & x \geq 1 \\ \log x & 0 \leq x < 1 \end{cases} \qquad f_p''(x) = \begin{cases} 2 & x \geq 1 \\ \frac{1}{x} & 0 < x < 1 \end{cases} \tag{31}$$

It's obvious that $f_p'(x)$ is continuous and $f_p''(x)$ is strictly positive. According to the second-order condition for convex functions, $f_p(x)$ is convex.

We then move on to verify the rest of the properties. It's obvious that given a specific $x > 0$, $f_p(x)$ is finite and $f_p(1) = 0$. In mathematics, the value of $x\log x$ when $x = 0$ is defined by its right limit, which means $f_p(0) = \lim_{x\to 0+} f_p(x) = 1$. $\qquad\square$

**Proposition 1**: *Given the piece-wise $f-$divergence in eq(30), $(f_p')^{-1}(x)$ and $f_p^*(x)$ has the following formulation:*

$$(f_p')^{-1}(x) = \begin{cases} \frac{x}{2} + 1 & x \geq 0 \\ e^x & x < 0 \end{cases} \qquad f_p^*(x) = \begin{cases} \frac{x^2}{4} + x & x \geq 0 \\ e^x - 1 & x < 0 \end{cases} \tag{32}$$

*Proof.* We begin with the proof of $(f_p')^{-1}(x)$. Given Lemma 2, the form of $(f_p')^{-1}(x)$ directly comes from the continuity and monotonic increasing property of $f_p'(x)$. One can easily verify $(f_p')^{-1}(x)$'s form by calculating the reverse function of $f_p'(x)$.

We then continue with the proof of $f_p^*(x)$. By the definition of Fenchel conjugate we have:

$$f_p^*(x) = \sup_{z\in[0,+\infty)} x \cdot z - f_p(z) \tag{33}$$

Because here $f_p(x)$ is a piecewise function, the supremum could be achieved in either $[0, 1)$ or $[1, +\infty)$. This means we should first take supremum over two intervals respectively and then choose the greater value as $f_p^*(x)$. More specifically, we perform the following decomposition:

$$f_p^*(x) = \max \left( \sup_{z \in [0,1)} x \cdot z - f_p(z), \sup_{z \in [1,+\infty)} x \cdot z - f_p(z) \right) \tag{34}$$

When constrained to either $[0, 1)$ or $[1, +\infty)$, $f_p(z)$ is no longer piecewise, and the inner supremum becomes the Fenchel conjugate, albeit under a narrower interval. To calculate Fenchel conjugate under a limited interval, one should observe that for any given $x$, the derivative of the inner objective satisfies:

$$\frac{d(x \cdot z - f_p(z))}{dz} = x - f_p'(z) \tag{35}$$

By Lemma 2, we have that $f_p'(z)$ is strictly increasing. This means $\frac{d(x \cdot z - f_p(z))}{dz}$ has only three possible behaviors on any interval: always positive, always negative, or transitioning from positive to negative with a zero crossing within the interval. Then for any interval $[a, b]$, $\sup_{z \in [a,b]} x \cdot z - f_p(z)$ has the following formulation. Note that similar results can be extended to open intervals.

$$f_p^*(x) = \begin{cases} x \cdot (f_p')^{-1}(x) - f_p((f_p')^{-1}(x)) & \text{if} \quad (f_p')^{-1}(x) \in [a, b] \\ \max \left( x \cdot a - f_p(a), x \cdot b - f_p(b) \right) & \text{else} \end{cases} \tag{36}$$

Leveraging this formulation, one can easily derive the close form solution of 34. In fact, $f_p^*(x)$ has the following formulation:

$$f_p^*(x) = \max \left( \begin{cases} \frac{x^2}{4} + x & x > 0 \\ x & x \leq 0 \end{cases}, \begin{cases} x & x \geq 0 \\ e^x - 1 & x < 0 \end{cases} \right) \tag{37}$$

The final form of $f_p^*(x)$ could be derived by directly taking the maximum. $\quad\square$

## C  In-depth Discussion of In-sample Guidance Learning and Other Guidance Methods

**Difference between IGL and other inexact guidance methods**    One important property of IGL is that it can induce the exact optimal policy, rather than a policy similar to the optimal policy, if the optimal policy can be represented as a weighted behavior policy. Some of the previous methods use inexact guidance terms and thus have no guarantee of the induced policy [17, 16, 6]. Among them, Janner et al. [17] uses a mean-square-error (MSE) objective to train the guidance model. Ho et al. [16], Chung et al. [6] leverage training-free guidance by reusing the pre-trained diffusion model in the data prediction formulation to characterize intermediate guidance.

**Difference between IGL and Contrastive Energy Prediction (CEP)**    Besides the inexact guidance methods mentioned above, Contrastive Energy Prediction (CEP) [32], is a guidance learning method that could induce exactly the desired policy if it can be represented as a weighted behavior policy. However, CEP is still prone to exploit overestimation error in the critic. We provide additional explanations for the differences between Contrastive Energy Prediction and In-sample Guidance Learning, as well as the reasons for these differences. Intuitively, the difference mainly stems from CEP's inherent property as a contrastive loss.

In order to obtain $\nabla_{a_t} \log \mathbb{E}_{p(a_0|a_t,s)}[w(s, a_0)]$, CEP will first train a model $g_\theta(s, a_t, t)$ to fit $\log \mathbb{E}_{p(a_0|a_t,s)}[w(s, a_0)]$ and then take the gradient of it. Its optimization objective has the following contrastive form:

$$\mathbb{E}_{P(t)} \mathbb{E}_{\pi^{\mathcal{D}}\left((a_0)^{(1:K)}|s\right)} \mathbb{E}_{p\left((a_t)^{(1:K)}|(a_0)^{(1:K)},s\right)} \left[ -\sum_{i=1}^K w\left(s, (a_0)^{(i)}\right) \log \frac{e^{g_\theta\left(s, (a_t)^{(i)}, t\right)}}{\sum_{j=1}^K e^{g_\theta\left(s, (a_t)^{(j)}, t\right)}} \right] \tag{38}$$

Note that here $K$ must satisfy $K > 1$ to ensure the optimal condition is $g_{\theta^*}(s, a_t, t) = \nabla_{a_t} \log \mathbb{E}_{p(a_0|a_t,s)}[w(s, a_0)]$. This means under offline setting, additional fake actions are required on every state in the dataset. CEP first uses diffusion model to pre-train a behavior policy $\hat{\pi}^{\mathcal{D}}$ from the

dataset and then generates multiple fake actions $(a_0)^{(1:K)} \sim \hat{\pi}^{\mathcal{D}}(a_0|s)$ for each state in the dataset. The second expectation in the objective will be calculated using these fake actions. It's clear that when $\hat{\pi}^{\mathcal{D}}$ mistakenly generates OOD actions, CEP still uses action evaluation model to predict $w(s, a_0^{OOD})$ on these OOD actions. The introduction of $w(s, a_0^{OOD})$ can make $\nabla_{a_t} \log \mathbb{E}_{p(a_0|a_t,s)}[w(s, a_0)]$ prefer these $a_0^{OOD}$. This is because $\nabla_{a_t} \log \mathbb{E}_{p(a_0|a_t,s)}[w(s, a_0)]$ will guide the samples towards actions with high $w(s, a_0)$ value. These high $w(s, a_0)$ value actions can be OOD actions whose $w(s, a_0)$ is overestimated. The catastrophic result is that CEP may guide action samples towards OOD actions with erroneously high $w(s, a_0)$ value rather than real high-quality actions.

Finally and most importantly, we will explain from a probabilistic perspective why the contrastive loss in CEP can't be estimated with one sample from $\pi^{\mathcal{D}}(a|s)$. The derivation of the optimality condition also depends on changing the conditional variables from $(a_0)^{(1:K)}$ to $(a_t)^{(1:K)}$:

$$\mathbb{E}_{P(t)} \mathbb{E}_{\pi^{\mathcal{D}}\left((a_0)^{(1:K)}|s\right)} \mathbb{E}_{p\left((a_t)^{(1:K)}|(a_0)^{(1:K)},s\right)} \left[ -\sum_{i=1}^{K} w\left(s, (a_0)^{(i)}\right) \log \frac{e^{g_\theta\left(s,(a_t)^{(i)},t\right)}}{\sum_{j=1}^{K} e^{g_\theta\left(s,(a_t)^{(j)},t\right)}} \right]$$

$$= \mathbb{E}_{P(t)} \mathbb{E}_{p\left((a_0)^{(1:K)},(a_t)^{(1:K)}|s\right)} \left[ -\sum_{i=1}^{K} w\left(s, (a_0)^{(i)}\right) \log \frac{e^{g_\theta\left(s,(a_t)^{(i)},t\right)}}{\sum_{j=1}^{K} e^{g_\theta\left(s,(a_t)^{(j)},t\right)}} \right]$$

$$= \mathbb{E}_{P(t)} \mathbb{E}_{p\left((a_t)^{(1:K)}|s\right)} \mathbb{E}_{p\left((a_0)^{(1:K)}|(a_t)^{(1:K)},s\right)} \left[ -\sum_{i=1}^{K} w\left(s, (a_0)^{(i)}\right) \log \frac{e^{g_\theta\left(s,(a_t)^{(i)},t\right)}}{\sum_{j=1}^{K} e^{g_\theta\left(s,(a_t)^{(j)},t\right)}} \right]$$
(39)

To derive the optimality condition that $g_\theta\left(s, (a_t)^{(i)}, t\right) = \log \mathbb{E}_{p\left((a_0)^{(i)}|(a_t)^{(i)},s\right)}\left[w\left(s, (a_0)^{(i)}\right)\right]$ for each $i$, one have to decompose the joint probability $p\left((a_0)^{(1:K)}|(a_t)^{(1:K)}, s\right)$ into the product of each $(a_0)^i$'s conditional probability:

$$= \mathbb{E}_{P(t)} \mathbb{E}_{p\left((a_t)^{(1:K)}|s\right)} \mathbb{E}_{\Pi_{i=1}^{K} p\left((a_0)^{(i)}|(a_t)^{(1:K)},s\right)} \left[ -\sum_{i=1}^{K} w\left(s, (a_0)^{(i)}\right) \log \frac{e^{g_\theta\left(s,(a_t)^{(i)},t\right)}}{\sum_{j=1}^{K} e^{g_\theta\left(s,(a_t)^{(j)},t\right)}} \right]$$

$$= \mathbb{E}_{P(t)} \mathbb{E}_{p\left((a_t)^{(1:K)}|s\right)} \left[ -\sum_{i=1}^{K} \mathbb{E}_{p\left((a_0)^{(i)}|(a_t)^{(i)},s\right)} w\left(s, (a_0)^{(i)}\right) \log \frac{e^{g_\theta\left(s,(a_t)^{(i)},t\right)}}{\sum_{j=1}^{K} e^{g_\theta\left(s,(a_t)^{(j)},t\right)}} \right]$$
(40)

Note that here we absorb the expectation over $\Pi_{i=1}^{K} p\left((a_0)^{(i)}|(a_t)^{(1:K)}, s\right)$ into the sum over $i$ and replace it with $p\left((a_0)^{(i)}|(a_t)^{(i)}, s\right)$. This is because each reverse process $p\left((a_0)^{(i)}|(a_t)^{(i)}, s\right)$ is independent of other $(a_t)^{(j \neq i)}$.

The requirement of decomposing joint probability $p\left((a_0)^{(1:K)}|(a_t)^{(1:K)}, s\right)$ means that $(a_0)^{(1:K)}$ should be mutually independent from each other, once $(a_t)^{(1:K)}$ and $s$ are determined. Obviously, if we only have one sample of $a_0$, $(a_0)^{(1:K)}$ must be identical and not independent. On the other hand, because the objective of In-sample Guidance Learning doesn't contain expectation over any joint distribution, it can be estimated with only one sample from $\pi^{\mathcal{D}}(a|s)$.

## D   Experimental Details

**Additional details for the practical algorithm**     In this part, we further provide the implementation details based on the given pseudo-code. Firstly, the score network $\epsilon_\theta$ we used is based on a U-net architecture, which is fairly common in diffusion-based RL algorithms [1, 59, 32]. For the DICE part, we use the double Q-learning trick [10] for the training of Q. For sampling, we use DPM-solver [31] to accelerate the sampling of the diffusion model, while the score function is given by IGL. The implementation of our score model and sampling process are based on DPM-solver, which uses MIT license. Diffusion-DICE contains 2 hyper-parameters in total, which are $\alpha$ in the DICE part and $K$ during sampling.

**Toy case experimental details**     As shown in Fig. 2, the behavior policy follows a standard Gaussian distribution constrained in an annular region. We sample 1500 times from the ground truth behavior policy to construct the offline dataset. The reward model $\hat{R}$ is represented with a 3-layer MLP and because we don't have to model complicated policy distribution, the score network $\epsilon_\theta$ in the toy case is simply multi-layer MLP. For better visualization, we sample 500 candidates during the *guide* stage. For QGPO, because it doesn't have the *select* stage, the sample distribution remains the same. For the other 2 methods, we select actions from candidates generated either from diffusion behavior policy (IDQL), or from guided policy (Diffusion-DICE) for 64 times. The reverse diffusion trajectories contain 16 action outputs. Because these 3 methods stand for different paradigms, we tuned different hyper-parameter sets for all of them to achieve the best performance.

**D4RL experimental details**     For all tasks, we ran Diffusion-DICE for $10^6$ steps and reported the final performance. In MuJoCo locomotion tasks, we computed the average mean returns over 10 evaluations across 5 different seeds. For AntMaze tasks, we calculated the average over 50 evaluations, also across 5 seeds. Because the inference of diffusion model is relatively slow, we conduct the evaluation every $4 \cdot 10^4$ training steps. Following previous research, we standardized the returns by dividing the difference in returns between the best and worst trajectories in MuJoCo tasks. In AntMaze tasks, we subtracted 1 from the rewards following previous methods [23, 34]. Note that for the D4RL benchmark and its corresponding datasets, Apache-2.0 license is used.

For the network, we use a 3-layer MLP with 256 hidden units to represent $Q$ and $V$. For the guidance network $g_\theta$, we use a slightly more complicated 4-layer MLP with 256 hidden units. Note that the time embedding in the guidance network is the Gaussian Fourier projection. We use Adam optimizer [21] to update all the networks, with a learning rate of $3 \cdot 10^{-4}$. The target network of $Q$ used for double Q-learning trick is soft updated with a weight of $5 \cdot 10^{-3}$.

For the baseline algorithms, we report the performance of IDQL under any number of hyper-parameters. Other baseline methods' results are sourced from their original papers. For all tasks, we list the chosen hyper-parameters in Table 2. As mentioned before, Diffusion-DICE requires significantly fewer action candidates during the select stage and we present the comparison of $K$ between Diffusion-DICE and IDQL in Table 3.

To further demonstrate the superior performance of Diffusion-DICE compared to traditional DICE algorithms, we conduct ablation studies by replacing the policy in Diffusion-DICE with a simple Gaussian policy, while preserving the piecewise $f$-divergence. The results are given in Figure 4.

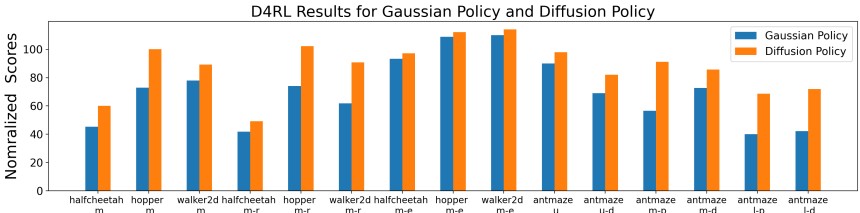

Figure 4: Although using piecewise $f$-divergence, the Gaussian policy in traditional DICE algorithm still results in inferior performance than using diffusion policy.

We also perform hyper-parameter study for candidate number $K$. To compare the performance of Diffusion-DICE under different $K$, we select some of the datasets in D4RL benchmark and test its performance following the same protocol before. Because the optimal $K$ for MuJoCo locomotion tasks and AntMaze navigation tasks have significant differences in magnitude, we chose to test 5 neighboring orders of magnitude around the optimal K value. The results are given in Table 4.

**Comparison of average $Q(s,a)$ curves**     For a detailed comparison between the *guide-then-select* paradigm and the *select-from-behavior* paradigm regarding error exploitation, we present their average $Q(s,a)$ curves normalized returns in Fig. 5. It's evident from the results that the average $Q(s,a)$ curves of the *select-from-behavior* paradigm almost remain above the *guide-then-select* paradigm, while the latter exhibits better performance.

**Learning curves on D4RL benchmark**     We present the learning curves on D4RL benchmark in Figure 6 and Figure 7.

Table 2: The chosen $\alpha$ and $K$ in Diffusion-DICE.

| Dataset | $\alpha$ | $K$ |
|---|---|---|
| halfcheetah-medium-v2 | 0.6 | 128 |
| hopper-medium-v2 | 0.6 | 32 |
| walker2d-medium-v2 | 0.6 | 32 |
| halfcheetah-medium-replay-v2 | 0.6 | 128 |
| hopper-medium-replay-v2 | 0.6 | 32 |
| walker2d-medium-replay-v2 | 0.6 | 32 |
| halfcheetah-medium-expert-v2 | 0.6 | 128 |
| hopper-medium-expert-v2 | 0.6 | 32 |
| walker2d-medium-expert-v2 | 0.6 | 32 |
| antmaze-umaze-v2 | 0.06 | 4 |
| antmaze-umaze-diverse-v2 | 0.8 | 1 |
| antmaze-medium-play-v2 | 0.06 | 8 |
| antmaze-medium-diverse-v2 | 0.06 | 8 |
| antmaze-large-play-v2 | 0.06 | 8 |
| antmaze-large-diverse-v2 | 0.06 | 8 |

Table 3: Comparison of $K$ between Diffusion-DICE and IDQL

| Dataset | D-DICE | IDQL |
|---|---|---|
| halfcheetah-medium-v2 | 32 | 256 |
| hopper-medium-v2 | 32 | 256 |
| walker2d-medium-v2 | 32 | 256 |
| halfcheetah-medium-replay-v2 | 32 | 256 |
| hopper-medium-replay-v2 | 32 | 256 |
| walker2d-medium-replay-v2 | 32 | 256 |
| halfcheetah-medium-expert-v2 | 32 | 256 |
| hopper-medium-expert-v2 | 32 | 256 |
| walker2d-medium-expert-v2 | 32 | 256 |
| antmaze-umaze-v2 | 4 | 32 |
| antmaze-umaze-diverse-v2 | 1 | 32 |
| antmaze-medium-play-v2 | 8 | 32 |
| antmaze-medium-diverse-v2 | 8 | 32 |
| antmaze-large-play-v2 | 8 | 32 |
| antmaze-large-diverse-v2 | 8 | 32 |

Table 4: Hyper-parameter Study for $K$

| $K$ | | 8 | 16 | 32 | 64 | 128 |
|---|---|---|---|---|---|---|
| | halfcheetah-medium-v2 | 48.8 | 52.2 | 55.6 | 58.4 | 60.0 |
| | hopper-medium-v2 | 89.6 | 96.8 | 100.2 | 98.9 | 96.1 |
| | walker2d-medium-v2 | 82.7 | 85.6 | 89.3 | 88.2 | 85.4 |
| Dataset | halfcheetah-medium-expert-v2 | 94.7 | 95.4 | 95.9 | 96.6 | 97.3 |
| | hopper-medium-expert-v2 | 109.3 | 110.6 | 112.2 | 111.9 | 110.8 |
| | walker2d-medium-v2 | 111.2 | 112.7 | 114.1 | 114.0 | 113.2 |

| $K$ | | 1 | 2 | 4 | 8 | 16 |
|---|---|---|---|---|---|---|
| | antmaze-umaze-diverse-v2 | 82.0 | 76.2 | 70.5 | 60.4 | 51.8 |
| | antmaze-medium-diverse-v2 | 78.8 | 81.2 | 83.3 | 85.7 | 82.9 |
| Dataset | antmaze-large-play-v2 | 55.8 | 61.0 | 65.3 | 68.6 | 67.2 |
| | antmaze-large-diverse-v2 | 60.2 | 66.8 | 70.1 | 72.0 | 69.9 |

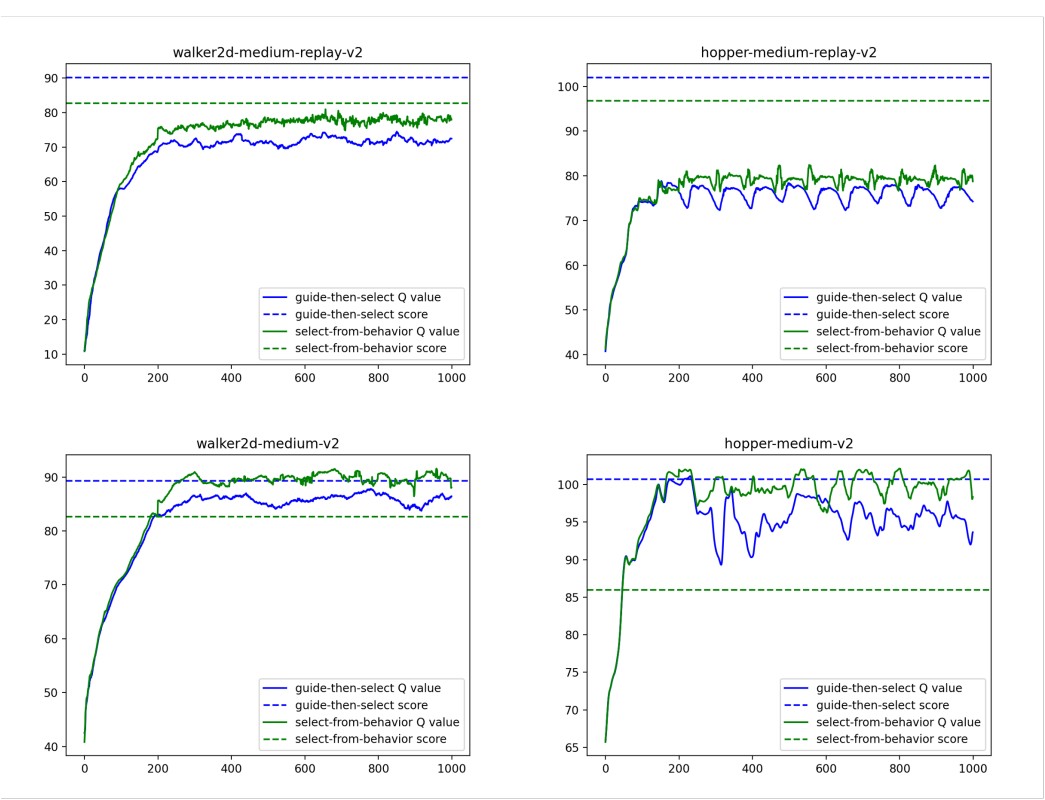

Figure 5: $Q(s, a)$ curves for *guide-then-select* paradigm and *select-from-behavior* paradigm.

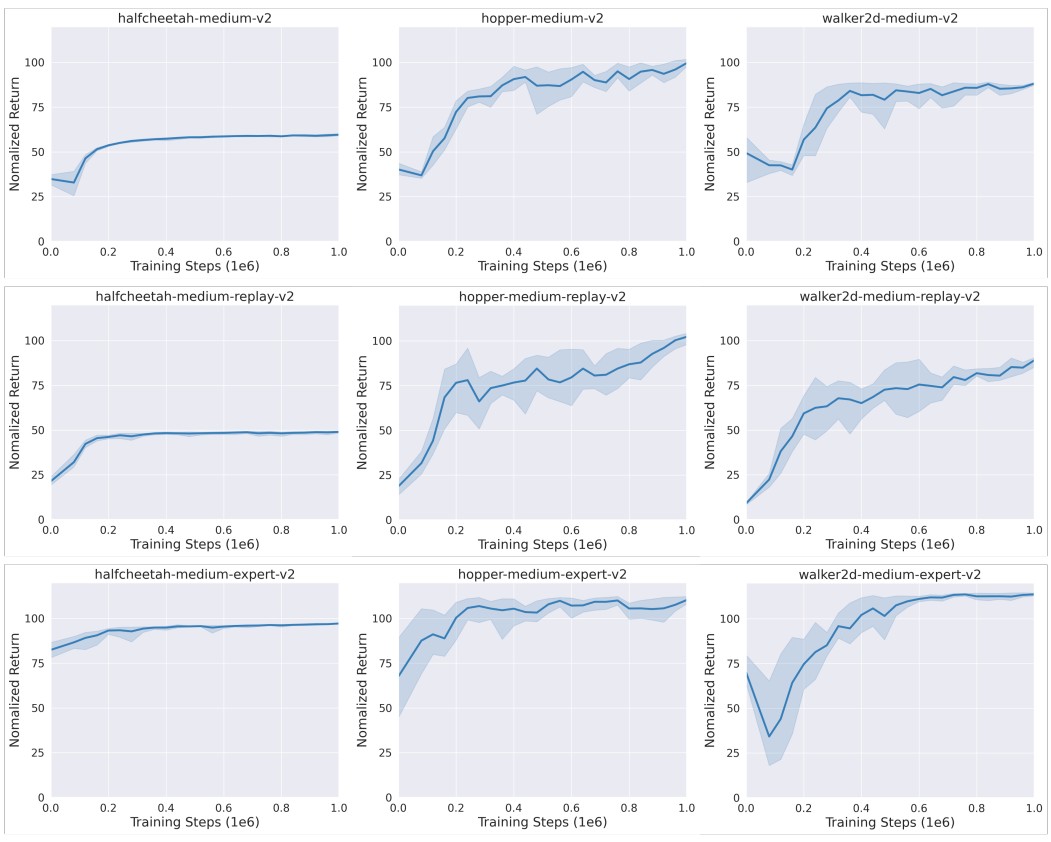

Figure 6: Learning curves of Diffusion-DICE on D4RL MuJoCo locomotion datasets.

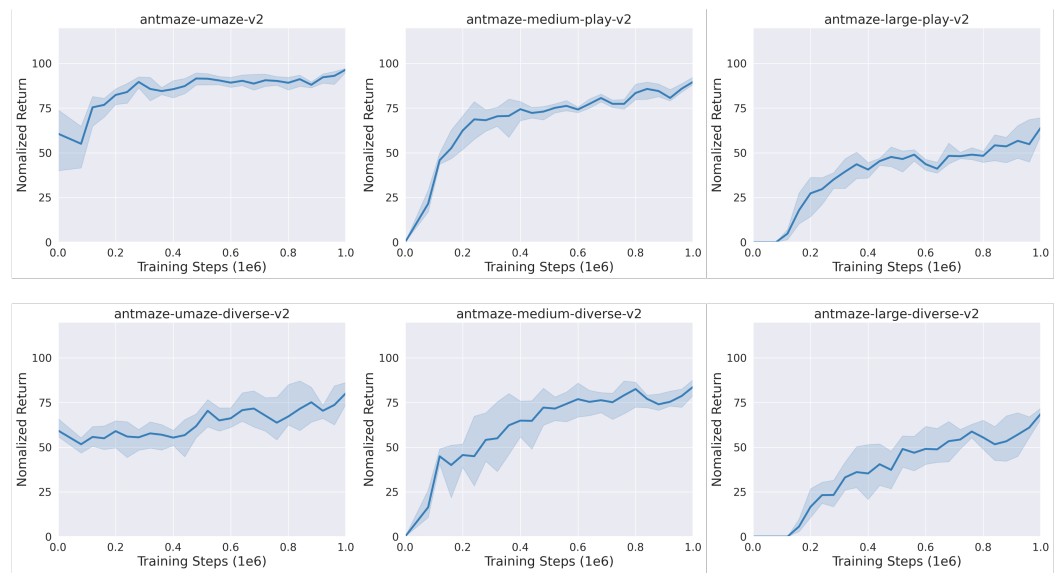

Figure 7: Learning curves of Diffusion-DICE on D4RL AntMaze navigation datasets.

# E    Experiments Compute Resources

For all the experiments, we evaluate Diffusion-DICE on either NVIDIA RTX 3080Ti GPUs or NVIDIA RTX 4090 GPUs. The training process of Diffusion-DICE takes about 6 hours on NVIDIA RTX 3080Ti or 4 hours on NVIDIA RTX 4090. The evaluation process of Diffusion-DICE takes about 3 hours on NVIDIA RTX 3080Ti or 2 hours on NVIDIA RTX 4090. Because we parallelize training and testing process, a complete experiment takes about 8 hours on NVIDIA RTX 3080Ti or 4 hours on NVIDIA RTX 4090.

# F    Broader Impacts

Besides the positive social impacts that Diffusion-DICE can help solve various practical offline RL tasks, for example, robotics, health care, industrial control tasks, some potential negative impacts also exists. Part of the Diffusion-DICE algorithm, the In-sample Guidance Learning, can also be applied to other generative tasks like image generation. This could be used to generate fake images, which could possibly mislead the public after being spread online.

