# OpenReview forum: "Diffusion-DICE: In-Sample Diffusion Guidance for Offline Reinforcement Learning"
_NeurIPS.cc/2024/Conference — NeurIPS 2024 poster_

### Official Review · Reviewer_UuBV · 2024-07-10

**Soundness:** 4
**Presentation:** 3
**Contribution:** 4
**Rating:** 8
**Confidence:** 4

**Summary:**

The paper mainly addresses the problem of obtaining the optimal policies in the distribution correction estimation (DICE) setting, which is one popular offline RL approach. Note that the offline RL assumes that we can’t interact with an environment and only have access to a dataset—sets of (state, action, and next state) tuples collected by some behavior policy. For the given history, DICE estimates the optimal value functions $V^*$ and $Q^*$ and obtains the optimal stationary distribution ratio $w^*$, which is the rate between the optimal policy’s state occupancy distribution $d^*$ and the one of the behavior policy $d^D$. Even if it is possible to get the optimal value functions and the ratio via DICE, obtaining the corresponding optimal policy $\pi^*$ from them is still challenging.

In this context, the paper proposes a novel training method for diffusion-based models to learn the optimal policy from the DICE-driven optimal value functions and the ratio. Unlike common density modeling, learning the diffusion-based models from the optimal value functions is not straightforward. Such a problem is closer to variational inference than density modeling; specifically, we don’t have samples from the target distribution to diffuse. Moreover, unlike a typical variational inference problem, sampling from the models is not available due to the nature of the offline RLs.

To do that the paper, the paper first introduces that the optimal policy can be represented by a product of the optimal stationary distribution ratio $w^*$ and the behavior policy $\pi^D$. Since the product of two distributions is unnormalized, the complete expression includes the normalization constant, which is the expected value of $w^*$ under the behavior policy (Line 155).

Next, the paper assumes that the behavior policy can be represented by a diffusion-based model with the forward process, which will also be used to perturb the policy of interest. Then, the authors show that the optimal policy at the perturbation level $t$ can be represented by using the behavior’s perturbed distribution at the same $t$ and the DICE ratio $w^*$ (Equation 6). In particular, this representation includes the logarithm of the expectation of the DICE ratio $w^*$ under the posterior distribution of the clean action $a_0$ for a given perturbed action $a_t$. While this novel representation doesn’t require sampling from the model policy anymore, computing the logarithm of the Monte Carlo estimate of any expectation is biased in general.

To circumvent this, the authors propose using the tangent transform, one of the variational approximation methods; thus, the quantity inside the logarithm is represented by an optimization problem, as in Equation 7. Interestingly, the new training objective only requires sampling from the behavior policy, which is suitable for the offline RLs.

In summary, the paper introduces a new representation of the diffusion-based optimal policy model by using the diffusion-based behavior policy and time-dependent learnable terms that will be learned by the convex problem described in Equation 7. The authors refer to this approach as In-sample Guidance Learning (IGL). While there have been a few previous approaches to learning diffusion-based optimal policy models, the authors point out that the IGL shows some benefits. For example, IGL only requires a single sample from the behavior policy, which can be obtained from the history, while some previous approaches require more than one sample from the behavior, which is not favorable in the offline RLs.

In addition, the authors suggest a few techniques to improve the stability of IGLs, such as using piecewise $f$-divergence for the DICE regularization term.

Finally, the paper discusses some failure cases of previous approaches and how the proposed method bypasses them. It also demonstrates the efficacy of the proposed method via experiments on several benchmark datasets.


-----------------------------------------------------
Updated the rating from 7 to 8 after the authors' rebuttal.

**Strengths:**

In my understanding, the paper's contributions are clear, and I also consider that the results are essential for several reasons.

The paper introduces a novel representation of the diffusion-based optimal policy model and its training method, IGL. In addition, the paper also motivates the solution well. For example, this approach circumvents the drawbacks of previous approaches, which require more than one sample from the behavior policy, and such actions may be out-of-distribution of the environments.

Moreover, the authors provide extensive discussions to help potential readers comprehend the characteristics of previous approaches and the proposed method.

Finally, the paper demonstrates the effectiveness of the proposed method via various experiments, which further supports the authors' claim.

**Weaknesses:**

Overall, the paper presents a novel method for learning optimal policy in DICE, which is a valuable contribution to the field. However, improvements in the presentation would greatly enhance the clarity and comprehensibility of the manuscript.

In particular, several equations within the paper omit the definitions of variables, which can lead to confusion—for example, $a_0$ in Line 119. In addition, some variables overlap while they are independent. For instance, in Line 155, the variable of the integration overlaps with the $a$ in the numerator.

I recommend revising the paper to address these issues.

**Questions:**

N/A

---

> ### Author Rebuttal · Authors · 2024-08-03
>
> We are deeply grateful for the reviewer's detailed and accurate summary. We also appreciate the time and effort the reviewer has devoted. As for the weakness, we prepared the following responses, which are presented as follows.
>
> >... However, improvements in the presentation would greatly enhance the clarity and comprehensibility of the manuscript.
>
> We apologize for any unclear expressions or improper organization in the article. We'll adjust our presentation in the updated version.
>
> >In particular, several equations within the paper omit the definitions of variables, which can lead to confusion—for example, $a_0$ in Line 119. In addition, some variables overlap while they are independent. For instance, in Line 155, the variable of the integration overlaps with the $a$ in the numerator.
>
> We apologize for omitting some definitions of variables and will include them in the updated version. In Line 119, $a_0$ represents the diffused action where the footprint stands for diffusion timestep. We'll also address the overlap of independent variables in the revised version.

---

### Official Review · Reviewer_uMN3 · 2024-07-11

**Soundness:** 3
**Presentation:** 4
**Contribution:** 3
**Rating:** 6
**Confidence:** 3

**Summary:**

The paper introduces a novel offline reinforcement learning approach that leverages diffusion models integrated with DICE-based methods. The proposed guide-then-select paradigm aims to minimize error exploitation. The resultant algorithm achieves state-of-the-art performance on D4RL benchmark tasks.

**Strengths:**

- Well-motivated and novel integration of diffusion models with DICE-based methods
- Well-written theoretical justification for the approach
- Strong empirical results that improve upon prior diffusion-policy baselines

**Weaknesses:**

- Missing comparison in Table 1 of more optimal Gaussian policy methods, e.g. EDAC [1]
- Selection of D4RL datasets is limited, e.g. what about expert/random datasets? Further environments like Adroit would also be interesting
- Discussion of hyperparameter choice in the Appendix is important and should be included in the experimental section.
- A comparison of the inference speed of Diffusion-DICE and prior baselines would be valuable.

Minor:
- Line 70: “M” -> “M=”
- Line 73: LP abbreviation not explained
- There is concurrent related work [2] which also performs an analogous transformation between the behavior distribution to an online policy with diffusion models for synthetic data generation.

[1] Uncertainty-Based Offline Reinforcement Learning with Diversified Q-Ensemble. Gaon An, Seungyong Moon, Jang-Hyun Kim, Hyun Oh Song. NeurIPS, 2021.

[2] Policy-Guided Diffusion. Matthew Thomas Jackson, Michael Tryfan Matthews, Cong Lu, Benjamin Ellis, Shimon Whiteson, Jakob Foerster. RL Conference, 2024.

**Questions:**

Please see the above weaknesses.

**Limitations:**

Limitations are discussed in the conclusion.

---

> ### Author Rebuttal · Authors · 2024-08-03
>
> We appreciate the reviewer's time and effort dedicated to evaluating our paper, as well as the constructive feedback provided. In response to the concerns and questions raised, we have prepared detailed answers, which are outlined separately below.
>
> >Missing comparison in Table 1 of more optimal Gaussian policy methods, e.g. EDAC [1]
>
> We list the results of EDAC and Diffusion-DICE in the following table. For EDAC's results, we copy the results on MuJoCo locomotion tasks from its original paper and copy the results on AntMaze navigation tasks from CORL[1]. It's evident from the results that on MuJoCo locomotion tasks, Diffusion-DICE is comparable to EDAC. While on AntMaze navigation tasks, EDAC totally fails[3]. We suggest that this is because such ensemble-based uncertainty estimation heavily depends on the dataset distribution and on AntMaze datasets, it's not reliable.
>
>
>
> |  | Diffusion-DICE | EDAC |
> | -------- | -------- | -------- |
> |    halfcheetah-m      |     60.0     |    65.9      |
> |    hopper-m      |     100.2     |    101.6      |
> |    walker2d-m      |    89.3      |     92.5     |
> |    halfcheetah-m-r      |     49.2     |    61.3      |
> |    hopper-m-r     |    102.3      |      101.0    |
> |    walker2d-m-r      |    90.8      |     87.1     |
> |    halfcheetah-m-e      |    97.3      |    106.3      |
> |    hopper-m-e      |    112.2      |     110.7     |
> |    walker2d-m-e      |    114.1      |    114.7      |
> |    antmaze-u      |    98.1      |      0.0    |
> |    antmaze-u-d      |     82.0      |       0.0   |
> |    antmaze-m-p      |    91.3      |       0.0   |
> |    antmaze-m-d      |    85.7      |       0.0   |
> |    antmaze-l-p      |    68.6      |       0.0   |
> |    antmaze-l-d  |   72.0   |   0.0   |
>
> >Selection of D4RL datasets is limited, e.g. what about expert/random datasets? Further environments like Adroit would also be interesting
>
> For the experiments of MuJoCo locomotion tasks, we only choose "medium", "medium-replay", "medium-expert" datasets because "random" dataset hardly exists in real-world tasks, and "expert" data is typically used for imitation learning settings. Furthermore, these datasets are also rarely used in other diffusion-based offline RL methods. To demonstrate Diffusion-DICE's superiority against other methods, we evaluate Diffusion-DICE on 2 tasks from kitchen and 2 tasks from adroit, following the same experimental setting in Appendix D. Note that we only choose 4 tasks in total due to the limited rebuttal preiod.
>
> |    | Diffusion-DICE | EDP | LD[2] | Diffusion-QL | QGPO|IQL|$f$-DVL|
> | -- | --- | - | --- | - | - |-|-|
> | kitchen-partial| **78.3** | 46.3 | -  | 60.5 | - | 46.3 | 70.0 |
> | kitchen-mixed |  **67.8**  | 56.5  | - | 62.6  | - |51.0| 53.8 |
> | pen-human | **84.4** | 72.7 | 79.0 | 72.8 | 73.9 | 71.5 | 67.1 |
> | pen-cloned  | **83.8** | 70.0| 60.7 | 57.3 | 54.2 | 37.3 | 38.1 |
>
> |      | $\alpha$ | K    |
> | ---- | ----- | ---- |
> |   kitchen-partial   |    0.6      |   4   |
> |    kitchen-mixed  |   0.6       |   4   |
> |   pen-human   |    0.6      |   4   |
> | pen-cloned | 0.6     | 8 |
>
> The results and the chosen hyperparameters are listed in the tables above. We compare Diffusion-DICE with other Offline RL baselines (either diffusion-based or not). The results are either taken from their original papers (if available) or from LD[2] (if not). The results on these more complex environments consistently show Diffusion-DICE's superiority.
>
> >Discussion of hyperparameter choice in the Appendix is important and should be included in the experimental section.
>
> We apologize that due to the page limit, the discussion of hyperparameter choice is placed in the appendix. In the updated version, we'll include this discussion in the experimental section.
>
> >A comparison of the inference speed of Diffusion-DICE and prior baselines would be valuable.
>
> In the following table, we compare the inference time (seconds/100 actions) of Diffusion-DICE and other baselines. It's worth noting that because we mainly focus on diffusion-based methods, these baselines also contain only diffusion-based algorithms. The results are based on a single RTX 4090 GPU, under antmaze-large-diverse-v2 environment.
>
>
>
> |       | Diffusion-DICE | SfBC |   QGPO  |  IDQL   |  Diffusion-QL   |  Diffuser   |
> | -- | --- | -- | -- | -- | -- | -- |
> | Inference Time (s/100 actions) |   16.86   |  20.12    |  15.50   |   6.32  |   1.65  |  98.64   |
>
> >Line 70: “M” -> “M=”
>
> We'll fix this typo in the updated version.
>
> >Line 73: LP abbreviation not explained
>
> We're sorry that due to the page limit, the explanation of LP abbreviation is omitted. In fact, we refer to LP as the expected return's linear programming form (linear with $d^\pi$). By the definition of $d^\pi(s, a)$ in Line 74, $d^\pi(s, a)$ represents the discounted sum of probability that the agent takes action $a$ on state $s$ over all steps $t$. Then it's obvious that $E_{(s, a) \sim d^\pi} [r(s, a)]$ equals to the discounted sum of rewards., i.e. $E[\sum_{t=0}^\infty \gamma^t \cdot r(s_t, a_t) ]$. Due to the bijection between $\pi$ and $d^\pi$, maximizing expected return over $\pi$ is equivalant to maximizing $E_{(s, a) \sim d^\pi} [r(s, a)]$ with respect to $d^\pi$. The latter exactly possesses a linear programming form.
>
> >There is concurrent related work [2] which also performs an analogous transformation between the behavior distribution to an online policy with diffusion models for synthetic data generation.
>
> Thanks for pointing out. We'll add this to the discussion in the updated version.
>
> [1]: CORL: Research-oriented Deep Offline Reinforcement Learning Library. Denis Tarasov, Alexander Nikulin, Dmitry Akimov, Vladislav Kurenkov, Sergey Kolesnikov. NeurIPS, 2024.
>
> [2]: Efficient Planning with Latent Diffusion. Wenhao Li. ICLR, 2024.
>
> [3]: Revisiting the Minimalist Approach to Offline
> Reinforcement Learning. Denis Tarasov, Vladislav Kurenkov, Alexander Nikulin, Sergey Kolesnikov. NeurIPS, 2023.

---

> > ### Author Response · Authors · 2024-08-12
> > **Kind Reminder: Discussion Period Ending**
> >
> > Dear Reviewer uMN3,
> >
> > We sincerely apologize for any inconvenience this reminder may cause. We just wanted to kindly remind you that the discussion period will conclude tomorrow.
> >
> > As the discussion period is nearing its end, we wanted to check if you have any remaining questions or concerns. **We would be more than happy to address further inquiries you may have.** We understand how busy you must be during this time, and we truly appreciate the effort and time you've dedicated to the rebuttal process.
> >
> > Thank you very much, and we look forward to your response.
> >
> > Best regards,
> >
> > The Authors of Paper 16567

---

> > ### Comment · Reviewer_uMN3 · 2024-08-13
> >
> > Thank you for your clarifications, I will raise my score.

---

### Official Review · Reviewer_QJqE · 2024-07-12

**Soundness:** 3
**Presentation:** 2
**Contribution:** 3
**Rating:** 6
**Confidence:** 3

**Summary:**

This paper introduces Diffusion-DICE for offline reinforcement learning. Diffusion-DICE motivates from the transformation between the behaviour distribution and the optimal distribution, which inspires the use of generative models for behaviour distribution modelling. Next, Diffusion-DICE decomposes the policy score function into two components, one from the behaviour distribution, another from the guidance, i.e., the transformation. Lastly, Diffusion-DICE employs a guide-then-select paradigm, which uses only in-sample actions for training to avoid out-of-distribution issues. In their experiments, Diffusion-DICE has achieved strong performance compared with baselines on the D4RL benchmark.

**Strengths:**

The proposed idea is novel and interesting. I like the way the authors connect DICE with diffusion policies, decompose the score functions, and make the guidance score tractable. Theoretically, they have provided careful analysis and derivations to support the claims. Empirically, they have conducted both toy experiments for intuitive understanding, and demonstrating the strong performance on the D4RL benchmark.

**Weaknesses:**

There are several weaknesses of the paper I’d like to point out.

1/ The biggest issue is the presentation. Although I do like the idea of the work and recognise its contributions, I found the paper very hard to follow and needed to read through the paper to understand the introduction. The way authors presented the guide-then-select is confusing. I’d suggest authors provide a bit more background and carefully define the “guidance term”, and how it relates to the RL before using it in both abstract and introduction.

2/ Achieving in-sample learning of offline diffusion RL is not new. Efficient Diffusion Policy (EDP) [1] has introduced an IQL-based variant which naturally allows training Q-values using only in-sample data, without querying out-of-distribution actions during policy evaluation. I’d suggest the authors carefully check the claims and avoid over claiming.

3/ The D4RL experiments are only conducted on the locomotion tasks and the antmaze tasks. The commonly tested kitchen and adroit tasks are missing, which weakens the claim of the paper.

References:

[1] Kang, B., Ma, X., Du, C., Pang, T., & Yan, S. (2024). Efficient diffusion policies for offline reinforcement learning. Advances in Neural Information Processing Systems, 36.

**Questions:**

Could you provide a bit more discussion about the differences of using guidance for inference with the Diffusion-QL style inference, which directly guides the sampling process towards actions with high returns? It would be interesting to understand the pros and cons of these two different paradigms

**Limitations:**

I think in general this is an interesting work and I don’t see major limitations.

---

> ### Author Rebuttal · Authors · 2024-08-03
>
> We appreciate the reviewer's time and effort dedicated to evaluating our paper, as well as the constructive feedback provided. In response to the concerns and questions raised, we have prepared detailed answers, which are outlined separately below.
>
> >... The way authors presented the guide-then-select is confusing. I’d suggest authors provide a bit more background and carefully define the “guidance term”, and how it relates to the RL before using it in both abstract and introduction.
>
> We apologize for the lack of explanation of 'guidance term' in the abstract and introduction due to the page limit. The 'guidance term' refers to the log-expectation term defined in Eq. (6), which 'guides' the diffused action towards high-value regions. In the updated version, we will provide more background information on this term and its relation to RL in the introduction.
>
> >Achieving in-sample learning of offline diffusion RL is not new. Efficient Diffusion Policy (EDP) has introduced an IQL-based variant which naturally allows training Q-values using only in-sample data, without querying out-of-distribution actions during policy evaluation. I’d suggest the authors carefully check the claims and avoid over claiming.
>
> We acknowledge that EDP is also a diffusion-based algorithm that avoids querying OOD actions' value, and we will add it to the discussion in the updated version. However, the major difference between EDP and Diffusion-DICE is that EDP has no guarantee of the form of optimal policy. According to EDP's original paper, it discusses two types of approaches: "direct policy optimization" and "likelihood-based policy optimization". It's obvious that the former can not guarantee the form of optimal policy. The latter replaces $\log \pi_\theta(a|s)$ with its lower bound in the optimization objective, which consequently loses guarantee for the form of optimal policy. On the other hand, Diffusion-DICE directly calculates the score function of the optimal policy induced from DICE's objective. As both terms in Eq. (6) can be estimated unbiasedly, the policy distribution induced by Diffusion-DICE can match the exact optimal policy distribution.
>
> >The D4RL experiments are only conducted on the locomotion tasks and the antmaze tasks. The commonly tested kitchen and adroit tasks are missing, which weakens the claim of the paper.
>
> To validate that Diffusion-DICE also demonstrates superior performance on other more complex tasks, we evaluate Diffusion-DICE in kitchen and adroit environments. Due to limited rebuttal period, we choose 2 tasks from kitchen and 2 from adroit. We compare Diffusion-DICE with other Offline RL baselines (either diffusion-based or not). The results are either copied from their original papers (if available) or LD[1] (if not). The results and the chosen hyperparameters are as follows:
>
> |                 | Diffusion-DICE | EDP | LD[1] | Diffusion-QL | QGPO|IQL|$f$-DVL|
> | ---------- | ------ | ------ | ------ | ------ | - |-|-|
> | kitchen-partial| **78.3** | 46.3 | -  | 60.5 | - | 46.3 | 70.0 |
> | kitchen-mixed |  **67.8**  | 56.5  | - | 62.6  | - |51.0| 53.8 |
> | pen-human | **84.4** | 72.7 | 79.0 | 72.8 | 73.9 | 71.5 | 67.1 |
> | pen-cloned  | **83.8** | 70.0| 60.7 | 57.3 | 54.2 | 37.3 | 38.1 |
>
> |      | $\alpha$ | K    |
> | ---- | -------- | ---- |
> |   kitchen-partial   |    0.6      |   4   |
> |    kitchen-mixed  |   0.6       |   4   |
> |   pen-human   |    0.6      |   4   |
> | pen-cloned | 0.6     | 8 |
>
> Note that we follow the same experimental settings in Appendix D. The results further substantiate our claim that Diffusion-DICE achieves optimal policy transformation while keeping minimal error exploited, and thus exhibits SOTA performance even on more complex tasks.
>
> >Could you provide a bit more discussion about the differences of using guidance for inference with the Diffusion-QL style inference, which directly guides the sampling process towards actions with high returns? It would be interesting to understand the pros and cons of these two different paradigms
>
> The major difference comes from the way score function is modeled. Diffusion-QL represents algorithms that directly model optimal policy's score function with one neural network. Diffusion-DICE represents algorithms that indirectly model it as a "transformed" score function of the behavior policy, with possibly more than one neural network.
>
> For Diffusion-QL, as the guidance towards high-value actions has already been encoded in the score network, simply running reverse diffusion process with the learned score network could bring high-value actions, which makes it easier to implement and faster to inference. However, as the marginal probability $\log \pi_\theta(a|s)$ for diffusion model is hard to compute, the policy improvement of such algorithms almost relies on surrogate objectives (See Eq. (3) in Diffusion-QL[2] and Eq. (10), Eq. (12) in EDP[1]). Consequently, the exact distribution of diffused action after inference is unknown, and the form of optimal policy is not guaranteed.
>
> On the other hand, using guidance for inference allows for the decoupled and exact learning of both the behavior policy's score function and the guidance term. This provides guarantee for the action distribution after inference. Moreover, due to the decoupling between behavior policy's score function and guidance term, it's possible to combine different guidance flexibly during inference, without training the desired score function from scratch. This is especially useful for aligning large diffusion models in the future. However, given a limited amount of data, using guidance for inference may introduce auxiliary models, which increase the computational burdens.
>
> [1]: Efficient Planning with Latent Diffusion. Wenhao Li. ICLR, 2024.
>
> [2]: Diffusion Policies as an Expressive Policy Class for Offline Reinforcement Learning. Zhendong Wang, Jonathan J Hunt, Mingyuan Zhou. ICLR, 2023.

---

> > ### Comment · Reviewer_QJqE · 2024-08-09
> >
> > I do appreciate the authors' efforts and detailed explanations. I feel most of my concerns are addressed. I still feel this is a good paper, although certain efforts are still needed for a better presentation. I'll keep my original score and vote for an acceptance.

---

### Decision · Program_Chairs · 2024-09-25

**Decision:**

Accept (poster)

**Comment:**

The reviewers found the proposed approach novel and interesting. The method was found theoretically sound and the experimental results convincing.  Questions mainly concerned the clarification of the presentation of the mehtod, and the need for better positioning with respect to existing offline RL methods.

The rebuttal focused on clarifying the contributions of this work with respect to related approaches, and provided aditional results on the kitchen and adroit environments, demonstrating the benefit of Diffusion-DICE as compared to the chosen panel of baselines. These additional clarifications were appreciated by the reviewers during the discussion phase, and a consensus was reached towards the acceptance of this work to NeurIPS 2024.